# Self-organized spatial targeting of contractile actomyosin rings for synthetic cell division

María Reverte-López [1], Nishu Kanwa[1], Yusuf Qutbuddin [1], Viktoriia Belousova [1], Marion Jasnin [2] & Petra Schwille [1]✉

A key challenge for bottom-up synthetic biology is engineering a minimal module for self-division of synthetic cells. Actin-based cytokinetic rings are considered a promising structure to produce the forces required for the controlled excision of cell-like compartments such as giant unilamellar vesicles (GUVs). Despite prior demonstrations of actin ring targeting to GUV membranes and myosin-induced constriction, large-scale vesicle deformation has been precluded due to the lacking spatial control of these contractile structures. Here we show the combined reconstitution of actomyosin rings and the bacterial MinDE protein system within GUVs. Incorporating this spatial positioning tool, able to induce active transport of membrane-attached diffusible molecules, yields self-organized equatorial assembly of actomyosin rings in vesicles. Remarkably, the synergistic effect of Min oscillations and the contractility of actomyosin bundles induces mid-vesicle deformations and vesicle blebbing. Our system showcases how functional machineries from various organisms may be combined in vitro, leading to the emergence of functionalities towards a synthetic division system.

Bottom-up synthetic biology is an interdisciplinary field currently fostering promising technological advancements to tackle the environmental and biomedical challenges of the future while it strives towards a fundamental goal: the construction of an artificial cell from a set of minimal functional modules[1–4]. As protocell models, the biomimetic chassis commonly used in the field are giant unilamellar vesicles (GUVs), membrane-enclosed containers capable of hosting biochemical reactions[5]. To ensure the autonomy and continuity of our artificial vesicular systems, key cellular features and processes must be recapitulated within protocells; particularly, their ability to divide and self-replicate, a critical step in a cell's life cycle[1,6].

In this regard, several strategies have been conceived to engineer a synthetic division module capable of mechanical membrane abscission[7]. While the reconstitution of well-characterized bacterial divisome machinery is a very promising approach[8,9], recapitulating division with a eukaryotic cytoskeleton-based toolbox is another intriguing strategy, due to the versatility and modularity of eukaryotic division proteins. Inspired by eukaryotic cytokinesis, the so-called engineering route works on assembling an actomyosin contractile ring at the GUV equator which, upon its controlled diameter reduction, is supposed to constrict the vesicle membrane until scission[10]. Two main cytoskeletal components are required for this in vitro ring assembly route: actin and myosin. Actin, in its filamentous form and together with its many regulatory binding proteins, assembles into bundles which positioned at mid-cell constitute the ring scaffold. To generate the tension required for ring constriction, actin must interact with many membrane-associated and scaffolding proteins like anillin, septins and the motor protein myosin. The interplay of these cytokinetic constituents and the contraction generated by myosin and other passive crosslinkers are presumably behind the contractility of the actin assembly and furrow ingression[11–13].

[1]Department of Cellular and Molecular Biophysics, Max Planck Institute of Biochemistry, Martinsried, Germany. [2]Helmholtz Pioneer Campus, Helmholtz Munich, Neuherberg, Germany; Department of Chemistry, Technical University of Munich, Garching, Germany. ✉e-mail: schwille@biochem.mpg.de

Several studies have shown the successful reconstitution of contractile actomyosin rings in vitro[14–16]. Of particular interest is the reconstitution inside GUVs by Litschel et al.[15]. Using talin-vinculin as bundlers, membrane-bound actomyosin rings induced transient deformation in vesicles. However, bundles slipped on the membrane and formed condensate clusters, impeding the radial targeting of contractile forces on the vesicle membrane. Indeed, a key requirement for the cytokinetic engineering route to succeed is the stable circumferential positioning of the contractile ring during constriction of the vesicle, ideally at mid-cell. To date, however, spatiotemporal control of actomyosin rings in GUVs has not been achieved. For the eukaryotic reconstitution approach this challenge is of particular complexity, since reconstitution of in vivo mechanisms to maintain mid-cell ring placement would require the synergistic integration of many proteins systems and signalling molecules, a currently unattainable feat[10].

Interestingly, recent studies have shown the robustness and versatility of a protein-based spatial positioning toolset: the MinDE system[17–19]. The *Escherichia coli* (*E. coli*) Min proteins are a reaction-diffusion system able to self-organize on membranes through ATP hydrolysis. Composed of three proteins—MinD, MinE and MinC—the Min complex has a particular function in vivo: the localization of the FtsZ division ring (Z-ring) in the middle of rod-shape bacteria. Their self-organization mechanism consists of three steps: (1) ATP-dependent dimerization of MinD promoting membrane attachment, (2) MinE recruitment to membrane-bound MinD stimulating MinD's ATP-ase activity, (3) ATP hydrolysis and detachment of the MinDE complex from the membrane. Following this mechanism of pattern formation, MinD and MinE oscillate from one pole of the cell to the other and inhibit the assembly of the Z-ring near the poles via depolymerization of FtsZ through MinC, the functional cargo protein that is not involved in MinDE self-assembly[20]. In vitro, however, a surprising functionality of the MinDE system was discovered[21]. When reconstituted on planar supported lipid bilayers and inside GUVs, the proteins MinD and MinE can non-specifically sort and position any membrane-bound cargo. More precisely, through a diffusiophoretic transport mechanism, MinD fluxes can spatiotemporally control molecules on the membrane by frictional forces and generate anti-correlated molecular patterns[17]. This biochemical function, although possibly irrelevant in vivo, could thus be exploited for the positioning of membrane-attached molecules and other biomimetic features in artificial systems[18,19].

In this study, we demonstrate the successful co-reconstitution of actomyosin architectures and the MinDE system inside GUVs. Upon optimization of encapsulation conditions, time-lapse imaging revealed the MinDE-driven diffusiophoretic positioning of actomyosin rings and bundle networks at mid-cell, where we observed equatorial furrow-like invaginations breaking spherical symmetry. Moreover, we show that, besides the spatiotemporal control of actomyosin bundles at the membrane, MinDE binding can induce bleb-like outward protrusions in single-phase vesicles and domain-specific deformations in phase-separated GUVs. Thus, the experimental insights here reported demonstrate that upon ATP hydrolysis, MinDE proteins not only aid in active contractile ring localization, but also generate mechanical work to remodel vesicle membranes during synthetic division. Overall, these results showcase the advantages of integrating synthetic toolsets of different origin to engineer artificial cells with advanced functionalities.

## Results

### Co-reconstitution of actomyosin networks and the MinDE system inside GUVs

To achieve the co-reconstitution of actomyosin networks and Min oscillations inside vesicles, we carried out encapsulation experiments via double emulsion transfer to identify optimal experimental conditions for the dynamic and functional interplay of both systems' components.

First, since G-actin and Min protein self-organization depend on factors like salt concentration, supply of ATP and the presence of divalent cations in solution, we tuned the inner environment of the vesicles to simultaneously facilitate actin filament polymerization and MinD dimerization, critical for MinD interaction with negatively charged amphiphiles and its cooperative binding to membranes. Given the importance of the membrane as catalytic matrix for the spatiotemporal organization of MinDE proteins, we generated vesicles containing negative charge in the bilayer to enable the self-organization of Min proteins into different oscillation modes[22,23]. In addition, we incorporated biotinylated lipids to link biotinylated actin filaments to the inner leaflet of the GUVs. Anchoring actin assemblies to the membrane through neutravidin-biotin bonds allowed us to exploit the diffusiophoretic capabilities of Min proteins, which require membrane-bound cargo to induce the ATP-driven transport of molecules on membranes[17].

Once optimal buffer conditions and membrane composition were identified, we chose fascin as the crosslinking protein to generate high-order actin bundle structures. As previously reported[15], by binding fascin-assembled bundles to the membrane via neutravidin-biotin bonds we obtained long and curved bundles that robustly bound to the membrane adapting to the vesicle curvature. Additionally, to accelerate actin polymerization kinetics and decrease MinDE wavelength and oscillation velocity, we employed Ficoll70 as macromolecular crowder[9,24,25], which also facilitated vesicle production. Finally, to make our actin assemblies contractile and render membrane deformations[14,15], the motor protein myosin II was added to the inner solution mix (Fig. 1a).

To investigate the effects of MinDE oscillations on the formation of actin-bundle architectures under crowding conditions, we encapsulated actin and fascin at different molar ratios together with myosin II in the presence and absence of Min proteins. To this end, we analyzed actin architecture types and quantified their frequency of appearance in terms of GUV sizes. Similar to recent cytoskeletal reconstitutions in GUVs[16,26], we observed four main actin phenotypes in both the presence and absence of Min proteins: soft bundle webs, actomyosin asters, flexible rings and stiff-straight bundles (Fig. 1b).

Although phenotype yields differed, Min oscillations supported the bundling and assembly of actomyosin architectures on GUV membranes. In particular, at a 0.25 fascin/actin molar ratio (Fig. 1c), when Min proteins were part of the reaction mix, we detected an increase in flexible ring yields, as well as a lower probability of stiff bundle formation, irrespective of vesicle size. As previously reported, the size of a spherically confining environment impacts actin-bundle architecture due to the persistence length of actin filaments and the spontaneous equatorial assembly of bundles to minimize their bending energy[14,26]. In both samples containing or in absence of Min proteins, the probability of flexible ring formation was significantly higher in small diameter vesicles (diameter $< 15\,\mu m$), whereas for medium and big vesicles the predominant phenotype was aster, reaching almost 50% formation probability in vesicles between $20–25\,\mu m$ in diameter and 80% for vesicles bigger than $25\,\mu m$. Similarly, when a 0.5 fascin/actin molar ratio was employed and a higher fascin/actin concentration was encapsulated, we observed the four aforementioned phenotypes in the presence and absence of Min proteins (Fig. 1c). Under these conditions, however, only the frequency of stiff bundle formation decreased upon addition of Min proteins and the probability of flexible ring formation drastically decreased for GUVs bigger than $15\,\mu m$ diameter in samples containing Min proteins or in their absence.

Furthermore, to study the evolution of this co-reconstituted system over time, we performed experiments in the presence of MinDE at 0.25 fascin/actin ratio and quantified the frequency of actin phenotypes observed on the sample at three timepoints: right after encapsulation, 7 h and 24 h after vesicle production (Supplementary Fig. 1a).

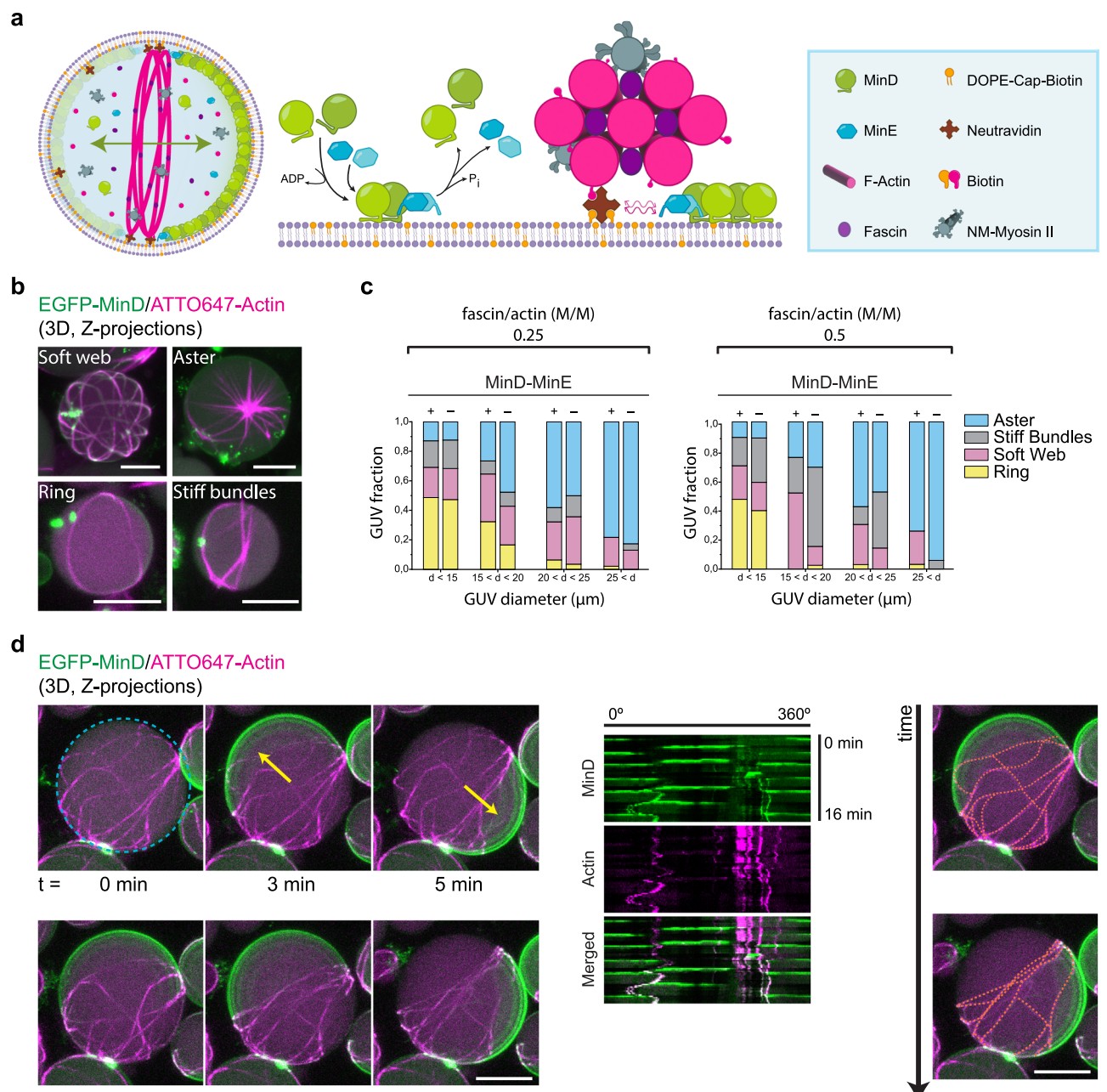

**Fig. 1 | Co-reconstitution of actomyosin networks and the MinDE system enables the reorganization and positioning of actomyosin bundles at mid-cell.** **a** Schematic illustration of the GUV content and the two macromolecular reactions at membrane level: the MinDE self-assembly mechanism behind pattern formation and the diffusiophoresis-mediated transport of neutravidin-bound actomyosin bundles by Min proteins. The active flux of MinDE proteins on the vesicle membrane interacts non-specifically via frictional forces with membrane-bound neutravidin inducing the transport and positioning of these molecules, and consequently the actomyosin bundles linked to them, towards areas of low MinD density. **b** 3D projections of confocal images showing the 4 phenotypes of actin architectures obtained after encapsulating 2.4 µM actin, 0.6 µM fascin (fascin/actin molar ratio = 0.25), 0.05 µM myosin II, 50 g/L Ficoll70, 3 µM MinD, 3 µM MinE and 5 mM ATP. Scale bars: 10 µm. **c** Bar graphs with the frequencies of the four

actomyosin phenotypes observed at different vesicle diameters when encapsulation experiments were performed at 0.25 and 0.5 fascin/actin molar (M/M) ratio in the presence and absence of Min proteins and protein/crowding conditions specified in (**b**). Total number of GUVs analyzed per condition = 150. **d** 3D projections of time-lapse confocal images depicting the reorganization and stacking of actomyosin bundles towards the vesicle equator driven by the diffusiophoretic transport of Min pole-to-pole oscillations. Yellow arrows indicate the perpendicular orientation of MinDE oscillations with respect to actomyosin bundles, which get antagonistically positioned at mid-cell. Kymographs generated at the vesicle equator (blue dashed circle) are meant to visually define the position of fluorescent features at this region over time. Orange dotted lines depict the approximate distribution of actin bundles on the membrane at two time points. Vesicle content as specified in (**b**). Scale bars: 10 µm. Source data are provided as a Source Data file.

Interestingly, we found that aster formation was immediate after vesicle generation, as the frequency of these star-like condensates remained markedly similar at 7 and 24 h: 34% and 38%, respectively. Conversely, vesicles initially presenting no distinguishable phenotype (but filled with G-actin on their lumen) showed a frequency as high as

asters right after encapsulation (40%), which progressively decreased over time reaching 6% at 24 h. Concomitantly, we observed that the number of vesicles containing rings and soft webs increased after 7 h, the latter phenotype showing progressive increment in number as we prolonged sample incubation time to 24 h.

Taken together, we show the successful co-reconstitution of the actomyosin system together with Min proteins and demonstrate that MinDE oscillations are compatible with the assembly of membrane-bound actomyosin architectures inside GUVs, and vice versa. Notably, addition of Min proteins promotes the formation of flexible acto-myosin rings in all vesicle sizes encapsulated with a 0.25 fascin/actin ratio.

## Diffusiophoresis-mediated positioning of actomyosin bundles at mid-cell by the MinDE system

Having established the conditions to reconstitute dynamic Min oscillations together with actomyosin-bundle assemblies inside GUVs, we then investigated whether Min proteins could effectively reorganize these assemblies and position them at mid-cell via their diffusiophoretic mechanism of molecular transport. Since flexible rings and soft bundle webs are the two types of actin architectures that could efficiently transmit contractile forces to the membrane, we studied the spatiotemporal organization of these two phenotypes by Min oscillations with time-lapse microscopy.

In agreement with past studies[27], we observed three main Min oscillation modes resultant from the reaction-diffusion fluxes of Min proteins on the inner leaflet of vesicles (Supplementary Fig. 1b): pulsing (oscillation characterized by the consecutive binding and unbinding of MinD to the entire vesicle membrane), pole-to-pole (sequential binding of MinD to the hemispheres of the vesicle), and circling waves (MinDE waves revolving around the inner leaflet of the membrane).

As MinDE pole-to-pole oscillations are the desired phenotype to actively transport molecules to the mid-cell region via diffusiophoresis[9,21], we first scrutinized actomyosin-containing vesicles exhibiting this dynamic pattern. Strikingly, in vesicles containing actomyosin bundles isotropically distributed all over the membrane, MinDE pole-to-pole oscillations yielded an anticorrelated and directional movement of the bundles perpendicular to the oscillation axis, reducing bundle interdistance and accumulating them at mid-cell (Fig. 1d, Supplementary Movie 1). Subjected to the highly dynamic MinDE pattern, the actomyosin bundles still showed positional fluctuations at the GUV equator over time, but maintained a perpendicular orientation to the oscillation axis.

Subsequently, to test the robustness of the MinDE diffusiophoretic transport in our actin-based encapsulation system, we varied the experimental conditions from our standard inner solution mix. We found that, in the absence of myosin II, under varying Ficoll70 concentrations (10-50 g/L), and employing different molar ratios for fascin/actin (0.25 or 0.5) as well as MinD/MinE ratios (1 or 2), Min proteins were capable of actively arranging membrane-bound actin bundles via diffusiophoresis when patterns different from pulsing developed on the vesicle membrane. Importantly, and as expected from our previous experiments, MinDE pole-to-pole patterns rotated and positioned fascin-assembled actin rings in the absence of myosin II, maintaining ring orientation perpendicular to MinDE oscillations (Supplementary Fig. 2b). In addition, when MinDE circling patterns emerged on vesicles containing fascin-bundled actin rings, we observed that the frictional forces induced by the directional MinD protein flux promoted the circular displacement of one end of the ring towards the opposite end (Supplementary Fig. 2c), resulting in the complete folding of the ring in less than 15 minutes (Supplementary Fig. 2d). Moreover, we detected that chaotic MinDE patterns—a dynamic mode in which Min proteins bind and unbind membrane areas in stochastic direction—could also alter the distribution of actomyosin bundles and, in some instances, buckle and collapse the network (Supplementary Figs. 1c, d).

Thus, our data demonstrate that the MinDE system can be used to regulate the spatiotemporal localization of membrane-bound actomyosin bundles and most importantly, position contractile actomyosin architectures at mid-cell via its characteristic pole-to-pole oscillation mode.

## Equatorial constriction of vesicles induced by positioned actomyosin architectures

In our system, myosin II not only acts as a crosslinking agent but also provides the contractile force required to induce membrane deformations. Having shown that the MinDE system localized actomyosin assemblies at the vesicle equator, we next investigated whether the positioned contractile assemblies could generate furrow-like membrane deformations.

First, to explore the contractile effect of myosin II on positioned actin structures, we carried out encapsulation experiments at 0.25 and 0.5 fascin/actin molar ratio with 50 g/L Ficoll70 and examined the vesicles presenting pole-to-pole oscillations with time-lapse confocal microscopy. Interestingly, when we employed a 0.5 fascin/actin molar ratio, mid-cell deformation induced by actomyosin ring constriction could be observed (Fig. 2a). This furrow-like invagination of the membrane, sustained over time (Supplementary Movie 2), generated two lobes where MinD proteins continued oscillating in a pole-to-pole pattern maintaining the localization of the ring at mid-cell.

In line with these observations, we also detected vesicle deformation at 0.25 fascin/actin molar ratio. Contrary to single rings, soft actomyosin bundle networks positioned at the vesicle equator formed a constriction band resulting in the loss of spherical vesicle shape (Fig. 2b). In addition to this large-scale membrane deformation, the vesicle presented an actin-filled membrane out-bud at the constriction site (Supplementary Fig. 3, Supplementary Movie 3). Although we could not acquire the formation process of this bud, Litschel et al. already reported on this type of membrane deformation, which results from the sliding of actomyosin bundles along the membrane to one side of the vesicle and their collapse into a condensate[15]. Notably, the membrane bud we observed was localized closely to the actomyosin band constricting the vesicle at the equator, while MinDE pole-to-pole oscillations continued at the two hemispheres generated at each side of the actomyosin band. To quantitatively assess vesicle deformation, we calculated the vesicle aspect ratio (AR) as the ratio between their major (a) and minor (b) axes. In both examples presented, aspect ratios indicated a vesicle deformation towards a rod-like shape (AR < 0.9), with the lowest AR obtained (0.75) 24 h after encapsulation, where a vesicle with a centered actomyosin network exhibited a distinct elongated shape (Supplementary Fig. 4a).

Furthermore, given that macromolecular crowders can impact both actin bundle architecture and the mechanical properties of the vesicle membrane[28,29], we subsequently performed encapsulation experiments at lower crowding concentration (20 g/l Ficoll70) and found membrane deformations in line with those already showed. Furthermore, under these experimental conditions, we detected an example of eccentric membrane deformation caused by an actomyosin ring when the established MinDE oscillation pattern inside the vesicle was different from the pole-to-pole mode (Fig. 2c). More specifically, the MinDE pattern at the vesicle membrane transitioned into a circling and less dynamic MinDE oscillation (possibly due to ATP depletion). Contrary to vesicles presenting mid-cell constriction, the membrane deformation observed induced a characteristic asymmetric dumbbell shape with two differently sized sub-compartments. Nevertheless, this asymmetric deformation, sustained over time (average AR over 20 minutes = 0.79), did not collapse after more than an hour of imaging (Supplementary Movie 4). Myosin II and ATP concentration added were 0.05 μM and 5 mM, respectively. Further experiments are therefore required to find the optimal encapsulating conditions enabling MinDE-stabilized rings positioned at mid-cell to undergo progressive contraction and controllably reduce their diameter by the action of myosin II motors.

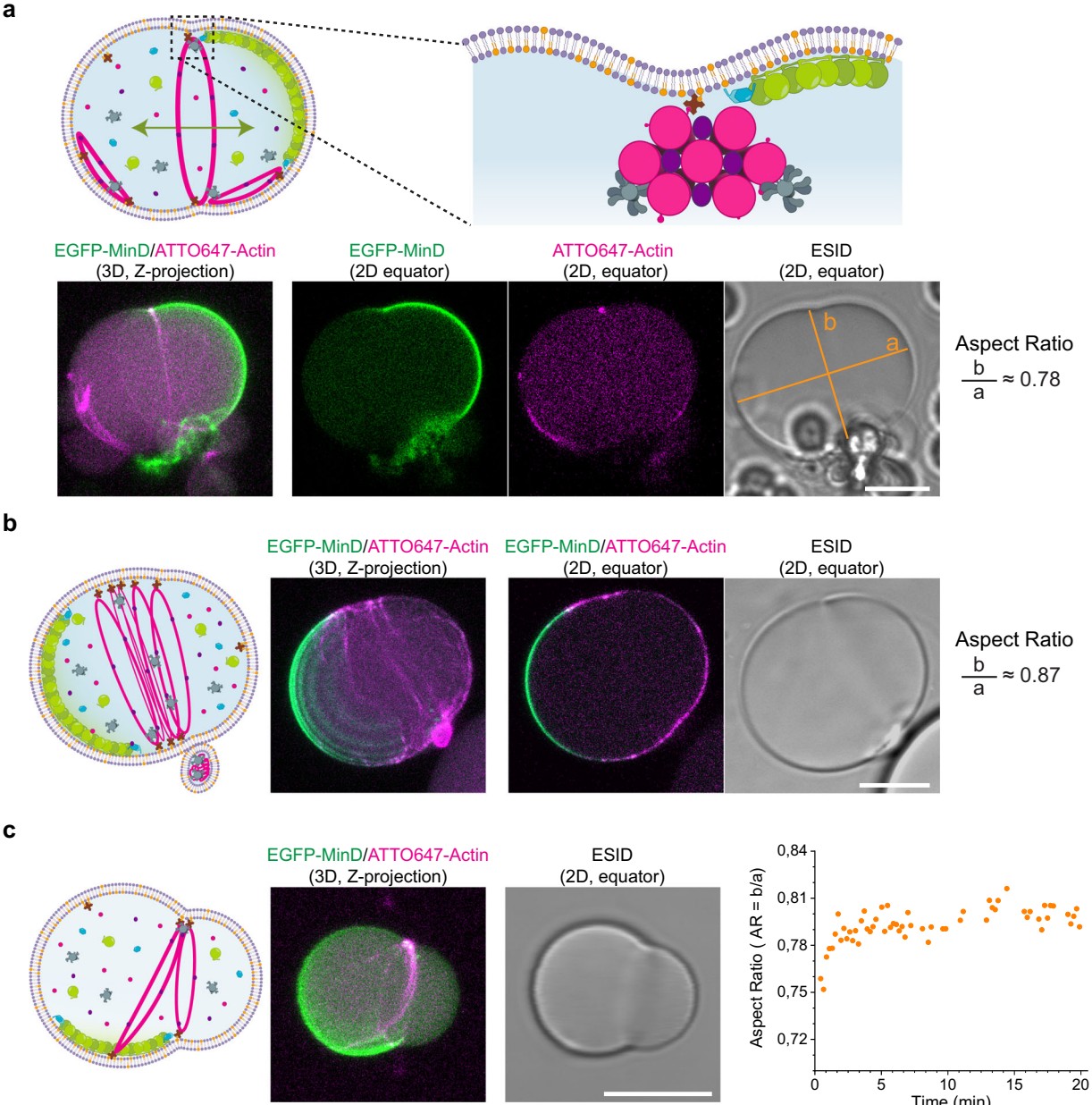

**Fig. 2 | Positioned actomyosin rings and soft webs constrict vesicles at mid-cell.** **a** Schematic illustration behind the mechanism of membrane deformation. Contractile actomyosin bundles positioned by MinDE proteins at mid-cell induce furrow-like membrane invaginations. 3D projections and 2D confocal images show an actomyosin ring constricting the vesicle at its equator. Orange lines indicate the major (a) and minor (b) axes measured to calculate the aspect ratio of the deformed vesicle (for spherical vesicles: aspect ratio = 1). Inner solution mix: 4 µM actin, 2 µM fascin (fascin/actin molar ratio = 0.5), 0.05 µM myosin II, 50 g/L Ficoll70, 3 µM MinD, 3 µM MinE and 5 mM ATP. Scale bar: 10 µm. **b** Schematic illustration, 3D projections and 2D confocal images of a vesicle containing a soft web of actomyosin bundles at the vesicle center being positioned by pole-to-pole Min oscillations. The contractile

actomyosin band formed causes the deformation of the vesicle (aspect ratio < 1). Inner solution mix: 2.4 µM actin, 0.6 µM fascin (fascin/actin molar ratio = 0.25), 0.05 µM myosin II, 50 g/L Ficoll70, 3 µM MinD, 3 µM MinE and 5 mM ATP. Scale bar: 10 µm. **c** Schematic illustration, 3D projection and 2D confocal image of a vesicle with a non-positioned contractile actomyosin assembly due to the loss in pole-to-pole MinDE oscillations. Constriction of the actomyosin bundles results in the deformation of the vesicle membrane into an asymmetric dumbbell shape. Scatter plot depicts the aspect ratio of the vesicle at different time points. Inner reaction mix: 4 µM actin, 2 µM fascin (fascin/actin molar ratio = 0.5), 0.05 µM myosin II, 20 g/L Ficoll70, 3 µM MinD, 3 µM MinE and 5 mM ATP. Scale bar: 10 µm. Source data are provided as a Source Data file.

In summary, we show that MinDE pole-to-pole oscillations can target the constriction of actomyosin architectures at the vesicle equator, resulting in the generation of sustained furrow-like membrane deformations.

## MinDE-induced blebbing in reconstituted actomyosin vesicles

When thicker and more abundant actomyosin bundles developed at the membrane in the form of soft webs, we detected the establishment

of more chaotic and static MinDE patterns. Thus, to further characterize the system, we performed time-lapse imaging on vesicles presenting chaotic or static-like MinDE patterns.

Interestingly, a large number of vesicles exhibiting these patterns developed membrane deformations similar to bleb-like morphologies (Fig. 3a, Supplementary Movie 5). To get further insights into the underlying mechanism behind bleb formation, we analyzed the interaction between our co-reconstituted protein systems and the vesicle

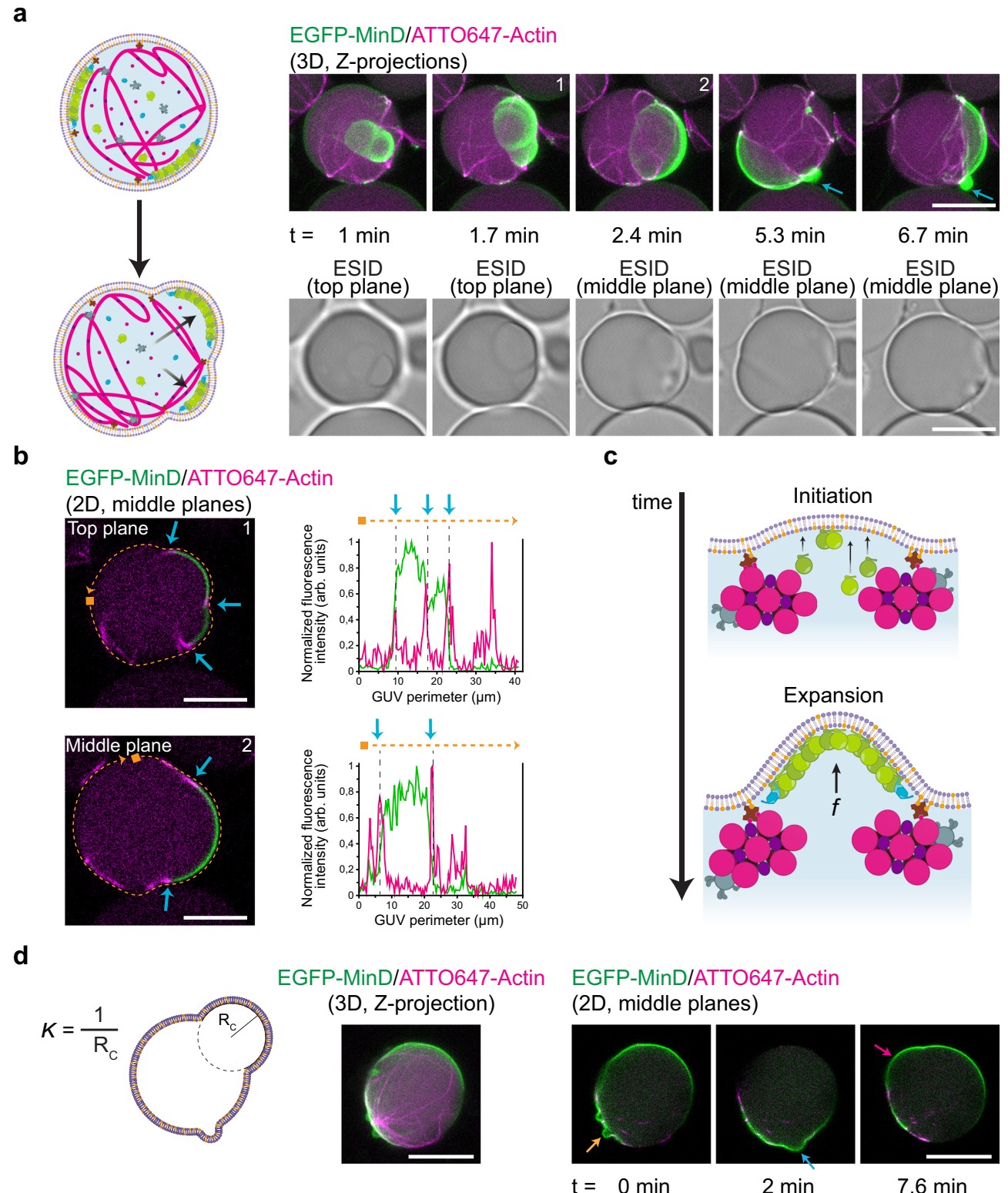

membrane. Closer inspection of the blebs' cross-sections revealed that the actomyosin soft web inside the vesicle compartmentalized the membrane into areas delimited by the peripheral attachment of bundles (Figs. 3b, Supplementary Fig. 5a). Due to this compartmentalization, container symmetry was lost and MinDE proteins generated a chaotic oscillation capable of deforming the membrane. More specifically, we found that the initial spontaneous curvature induced on these compartments by MinDE binding increased as more MinD molecules were recruited to the membrane, resulting in the outward

growth of the dynamic bleb-like protrusions (Fig. 3c, Supplementary Fig. 5b). Subsequently, after MinDE-driven membrane deformations, we observed that the reduction in bilayer tension and the recovery of the initial vesicle shape was accompanied by the generation of a membrane out-bud (Fig. 3a, blue arrows), which was not reabsorbed into the mother vesicle (Supplementary Fig. 5c)[30].

Furthermore, consistent with our previous observations and simultaneous to this membrane remodelling effect, the diffusiophoretic transport of actomyosin bundles reorganized the network at

**Fig. 3 | MinDE-induced blebbing in vesicles containing reconstituted acto-myosin architectures. a** Schematic illustration depicting the change in vesicle shape due to MinDE chaotic oscillations. Min proteins attach to areas delimited by soft actomyosin bundles and deform the membrane generating dynamic bleb-like protrusions. Fluorescence and brightfield confocal time-series show a blebbing vesicle. After bleb retraction, the reduction in bilayer tension generates an outward lipid bud (blue arrows). Encapsulation conditions: 2.4 μM actin, 0.6 μM fascin, 0.05 μM myosin II, 50 g/L Ficoll70, 3 μM MinD, 3 μM MinE and 5 mM ATP. Scale bars: 10 μm. **b** Confocal cross-section images at two time points of the vesicle in section a. Peripheral actomyosin anchoring creates a delimiting area which deforms upon MinDE binding. Additionally, MinDE diffusiophoretic transport changes the position of actomyosin bundles and the shape of the membrane area available for Min protein recruitment (blue arrows). Fluorescence intensity line plots of EGFP-MinD (green) and ATTO647-actin (magenta) demonstrate the demixing of both protein systems at the membrane perimeter (orange dotted line). Scale bars: 10 μm. **c** Schematic illustration of the proposed mechanism behind MinDE-induced blebbing. The recruitment of MinDE proteins to the compartmentalized inner leaflet of the bilayer generates the effect of a membrane outward protrusion in bleb form. **d** Schematic illustration depicting the radius of curvature $R_C$ used to calculate the curvature ($K = 1/R_C$) of the blebs. 3D projection and 2D time-lapse confocal images show a vesicle with diverse bleb-like deformations emerging over time. Orange arrow points at a bleb with $K = 0.73\ \mu m^{-1}$. Blue arrow, $K = 0.27\ \mu m^{-1}$. Magenta arrow, $K = 0.10\ \mu m^{-1}$. Encapsulation mix: 4 μM actin, 2 μM fascin, 0.05 μM myosin II, 50 g/L Ficoll70, 3 μM MinD, 3 μM MinE and 5 mM ATP. Scale bars: 20 μm. Source data are provided as a Source Data file.

the membrane. As a result, membrane compartments changed in size and the oscillations maintained a chaotic mode inducing dynamic blebs in other areas of the vesicle (Fig. 3b).

To further scrutinize the membrane-remodelling capabilities of Min proteins along with our actomyosin architectures, we performed encapsulation experiments at 0.5 fascin/actin molar ratio and calculated the curvature of the blebs observed. Similar to our non-deflated vesicles encapsulated with 0.25 fascin/actin ratio, MinDE binding induced blebbing in a subset of vesicles (Supplementary Movie 6). Time-lapse imaging revealed that Min proteins can induce blebs with a wide range of curvatures ($K$, calculated as the inverse of the radius of a circle that fits the bleb). As MinDE established a chaotic oscillation inside the vesicle, small blebs ($K = 0.73\ \mu m^{-1}$), medium size ($K = 0.27\ \mu m^{-1}$) and big ($K = 0.10\ \mu m^{-1}$) deformations emerged (Fig. 3d, orange, blue, and magenta arrows, respectively).

Hence, our results show that, when co-reconstituted with actomyosin bundle networks, the MinDE system can generate dynamic bleb-like outward protrusions in vesicles encapsulated at iso-osmolar conditions, confirming its capabilities as a membrane remodelling protein system as previously observed[27,31].

## Co-reconstitution of actomyosin bundle networks and the MinDE system in phase-separated vesicles show remodelling of membrane domains

Lastly, to increase the complexity of the system and test its compatibility with other shape-remodelling strategies, we first tested its reconstitution in vesicles of ternary lipid mixtures. Due to their tuneable mechanical and biochemical properties, phase-separated lipid membranes constitute another approach to aid in the remodelling of biomimetic systems by altering membrane curvature, fluidity, etc[32–34]. Furthermore, two-phase vesicles constitute an additional strategy to study the reorganization and deformation of free-standing lipid domains by actomyosin networks[35–37].

To achieve the co-reconstitution of the two protein systems in GUVs with phase-separated lipid domains we again employed the double emulsion transfer method. At room temperature (25 °C), GUVs demixed into coexisting Liquid-ordered (Lo) and Liquid-disordered (Ld) domains, where Ld domains consisted of DOPE-Biotin to facilitate actin binding (Fig. 4a). To study the successful reconstitution of the system inside phase-separated vesicles, we again performed time-lapse confocal imaging and observed that Min proteins could oscillate by binding to Ld domains on the vesicle membrane. Notably, due to the high frequency of soft actomyosin bundle webs formed, MinDE proteins also established chaotic oscillations on the Ld domains of the vesicle. The flexible bundles, however, spanned and crossed both lipid domains.

Interestingly, MinDE-driven diffusiophoretic transport displaced the actomyosin bundles bound at Ld/Lo boundaries, reorganizing the actomyosin network inside the vesicle. Moreover, consistent with our previous experiments performed on single-phase GUVs, MinDE binding to areas delimited by actomyosin bundles at domain boundaries deformed Ld domains into dynamic bleb-like protrusions (Figs. 4b, Supplementary Movie 7). In contrast to studies in which bulging or budding of phase-separated GUVs is externally induced by tuning membrane composition or changing osmotic conditions, and where deformations arise by an imbalance between surface tension and interfacial line tension[34,38], our phase-separated GUVs remained spherical over time in the absence of Min proteins, and no deformations in the form of blebbing or budding were observed (Supplementary Figs. 6, Supplementary Movie 8). Only in the presence of MinDE oscillations in GUVs containing actomyosin networks, timelapses showed the active—i.e., energy-consuming—deformation of Ld domains into outward blebs (Fig. 4c). During the course of these transient outward deformations, we also observed the remodelling of domains at the membrane such as their maneuvering followed by splitting (Supplementary Movie 7). As a result of MinD binding to Ld domains, the initial demixing of lipid phases on the vesicle thus changed, and domain reorganization occurred on the entire vesicle. Interestingly, while the resultant dynamic protrusions in the membrane comprised of Ld domains, Lo domains stayed intact.

In addition, to test the compatibility of our system with other membrane-based strategies for the enhancement of GUV deformation, we generated vesicles containing DOPE, a lipid with intrinsic negative curvature due to its inverse-cone shape[39]. Although the yield of vesicle production dramatically decreases at 15% DOPE molar ratio, successful encapsulation of our co-reconstituted system is possible when 10% DOPE is added to the lipid mix together with 60% POPC and 30% POPG. Importantly, in vesicles generated under these conditions, we observed generation of the three standard Min oscillations, actomyosin-driven equatorial deformation of vesicles and the formation of highly protruding blebs due to MinDE chaotic oscillations (Supplementary Fig. 4b).

In conclusion, we demonstrate that the co-reconstitution of Min proteins and actomyosin can deform DOPE-containing and complex ternary vesicle membranes, and dynamically remodel phase-separated lipid domains by rearranging and reshaping them on the membrane.

## Discussion

In this study, we have successfully co-reconstituted contractile actomyosin rings and other assemblies with a protein-based spatial positioning toolset, the MinDE system. Our results demonstrate that, under optimal encapsulating conditions, actomyosin bundles can be spatiotemporally controlled at the membrane and positioned at mid-cell via pole-to-pole MinDE oscillations. Notably, the positioned bundles, due to their contractile nature, induced membrane deformations, breaking GUV spherical symmetry and generating furrow-like invaginations. In addition to the observed actomyosin-driven membrane remodelling, we provide direct evidence that MinDE proteins generate dramatic bleb-like deformations upon dynamic binding to the membrane, another source of symmetry breaking for the GUVs. Thus, our findings lend further credence to the hypothesis that the diffusiophoretic function of the MinDE system can be exploited to maintain actomyosin

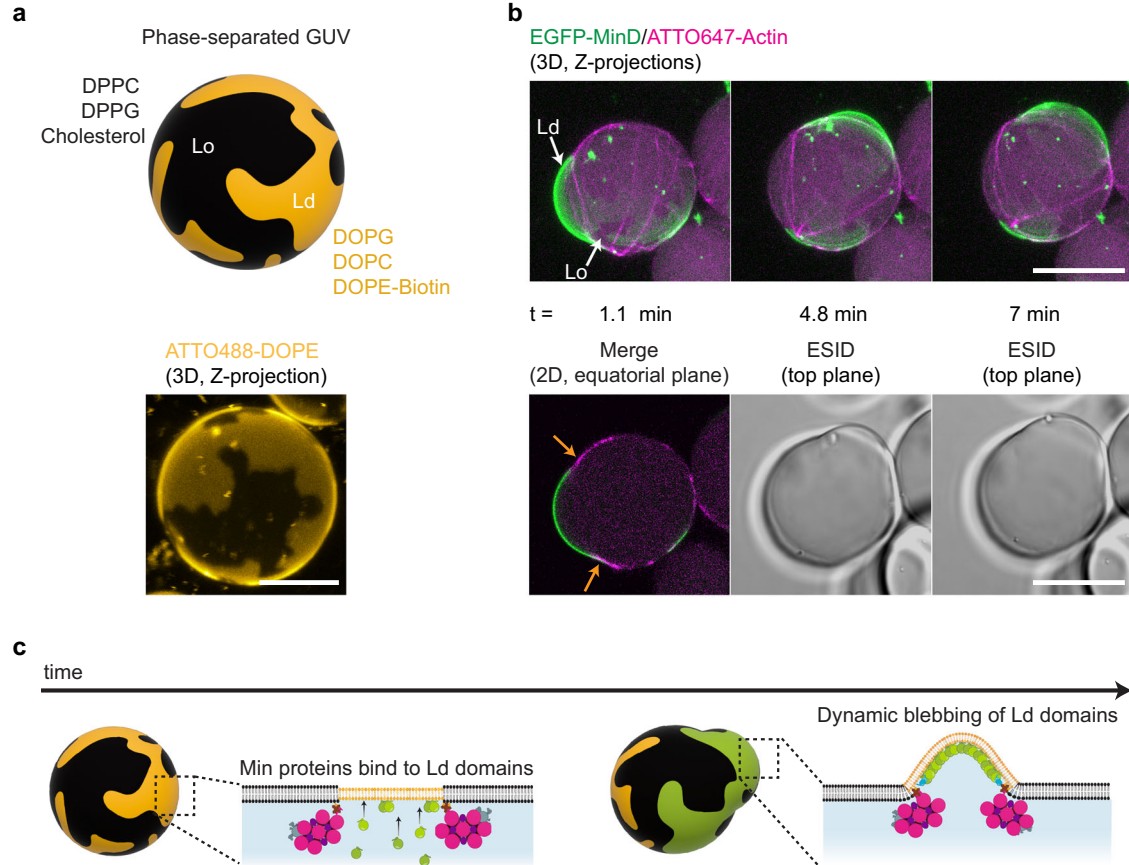

**Fig. 4 | MinDE-induced bleb morphologies on phase-separated GUVs with encapsulated actomyosin architectures. a** Schematic illustration (top) and 3D confocal image (bottom) show the membrane composition employed to generate phase-separated vesicles and the domains obtained. Scale bar: 10 μm. **b** 3D projections and 2D confocal images depict a blebbing phase-separated vesicle. MinDE proteins bind and oscillate on Ld domains. Actomyosin bundles remain at lipid-phase boundaries as Min proteins transiently deform Ld domains (orange arrows). Inner encapsulation mix: 2.4 μM actin, 0.6 μM fascin (fascin/actin molar ratio = 0.25), 0.05 μM myosin II, 20 g/L Ficoll70, 3 μM MinD, 3 μM MinE and 5 mM ATP. Scale bars: 20 μm. **c** Schematic illustration of the proposed mechanism behind the dynamic deformation of Ld domains by MinDE protein oscillations.

ring localization at mid-cell, one of the key milestones so far not successfully reported in the eukaryotic-based approach to synthetic cell division.

It should be noted that, after the adjustment in encapsulating conditions, our reconstitution conclusively showed the robustness of the MinDE system when integrated with the contractile actomyosin toolset. Not only did we observe the main MinDE-oscillation phenotypes extensively reported in past studies[9,27], our results also indicate that the establishment of Min patterns on the membrane had no detrimental effect of the formation of lipid-linked actomyosin structures. Indeed, we observed an increase in the frequency of GUVs containing membrane-bound rings when Mins were part of the inner encapsulation mix. Moreover, in agreement with previous MinDE geometry-sensing studies and the spatiotemporal feedback observed with FtsZ[40,41], our results demonstrate that Min oscillations orient themselves in a pole-to-pole fashion perpendicular to the spatially positioned ring structures while vesicles acquired ellipsoidal shape due to actomyosin contraction, an important aspect to consider as constriction progresses.

Alternative strategies for ring localization, like the use of microfluidic traps and curvature inducing/sensing biomolecules (e.g., septins, BAR domains, DNA origami), propose the use of external mechanical forces or biomolecules to generate a furrow-like negative curvature that could favor ring assembly[42–46]. However, due to progressive membrane deformation, ring alignment could be lost[42]. The

MinDE system, by contrast, dynamically responds to vesicle shape changes, as MinDE pole-to-pole oscillations block any membrane binding on the poles, thereby enabling the targeting of mechanical forces at the center[9]. Nonetheless, to keep the actomyosin ring in place and prevent slippage of the bundles on the membrane, future research should investigate the co-reconstitution of scaffold (e.g., anillin, septins) and severing/de-polymerizing proteins to increase turnover dynamics within actin bundles at the membrane as myosin contraction remodels the ring[47,48].

Besides accomplishing the active positioning of actomyosin architectures, time-lapse imaging revealed a surprising finding, the MinDE-driven blebbing of vesicles. In vivo, blebbing occurs at the cell poles as a mechanism to reduce cortical tension and ensure the stability of cellular shape during eukaryotic cytokinesis[49]. In our in vitro reconstitution, however, blebs originate because of a chemo-mechanical force induced by MinDE on free-standing membranes and constitute a source of asymmetric remodelling of the membrane[50,51].

Interestingly, whereas previous studies on Min proteins enclosed in GUVs have shown shape fluctuations in vesicles due to dynamic MinDE pulsing patterns under hypertonic and (near)-isotonic conditions[27,31,52], our results reveal MinDE-driven deformations of localized membrane regions which bend away from the inner leaflet where Min proteins attach. This observation leads us to speculate that the dramatic bleb-like protrusions shown here are due to confinement

effects induced by actomyosin bundles acting as lateral diffusion barriers for Min protein fluxes[53]. Consequently, and in accordance with previous studies, this partitioning of the vesicle membrane would in turn alter the membrane-to-bulk ratio and thus MinDE membrane kinetics[9,40]. The exact structural mechanism by which Min protein dynamics may induce local membrane deformations still remains opaque. Although previous reports have indicated the ability of Min proteins to increase membrane viscosity and change membrane topology by inserting their alpha helix[54,55], these two effects would not explain the outward budding force observed here[30,56]. Furthermore, while MinD-ATP forms flexible 2D oligomers on membranes, possible scaffolding effects due to their clustering at the inner leaflet remain unknown, and a potential intrinsic curvature of these oligomers is yet to be elucidated[57–59].

However, numerous theoretical and in vitro studies have already reported on the forces induced by reaction-diffusion systems such as Min proteins to deform open or confined dynamic surfaces[31,51,60,61]. More specifically, Fu et al. showed that only in the presence of MinE and ATP, MinDE oscillations could generate up to ~0.84 pN forces resulting in membrane extension and spreading of flat vesicles[31]. In contrast to other non-oscillatory protein-binding systems, which were shown to induce an opposite bending curvature due to protein-crowding effects[56,62], our MinDE protein system presents non-equilibrium dynamics at the inner leaflet and oscillatory membrane-bulk exchange due to ATP consumption. Therefore, to explain the outward membrane bending in our blebbing vesicles, we hypothesize that reactive MinDE fluxes localized to bundle-delimited regions exert an osmotic pressure arising from ATP-driven forces with normal components (Fig. 3c)[53,63]. In particular, the collisions resulting from the dynamic and highly cooperative binding and disassociation of MinD and MinE on the membrane patches, and the protein gradients generated on the proximal bulk, could first drive the straightening of nanoscale membrane undulations to release new membrane area, and ultimately, lead to outward blebbing[54,64].

Nevertheless, as some phenomenological aspects of the MinDE-membrane dynamics are still under investigation[58,65], future theoretical studies using modelling approaches combining reaction-diffusion dynamics and deformable surfaces should provide further insights to analyze this non-equilibrium process and underpin the precise molecular mechanism by which MinDE reactive fluxes are translated into shape-remodelling forces.

In addition, we observed an interesting mechanical effect on actin bundles themselves, arising via diffusiophoresis from MinDE dynamic patterns like chaotic or circling oscillations. Interestingly, time-lapse imaging showed bundles bending, collapsing, and stretching due to the diffusiophoretic frictional force of MinD fluxes. Thus, the system here reconstituted presents itself as another approach to study the crosslinker-dependent mechanical properties of actomyosin bundles and investigate the extent of diffusiophoresis when stiff supramolecular structures are bound to the membrane.

Importantly, and in accordance with previous studies, we observed actomyosin-driven deformation of GUVs under (near)-isosmotic conditions towards both ellipsoidal and asymmetric lobe shapes[15,16]. In this context, Bashirzadeh et al. calculated the force (~176 pN) generated by actomyosin rings on GUVs to induce asymmetric dumbbell deformations similar to those we observed (Fig. 2c)[16]. In contrast to bleb-like morphologies (Figs. 3 and 4), where forces arising from Min oscillations were essential to generate these deformations (Supplementary Fig. 6), in this study we also showed vesicles that exhibited sustained equatorial constriction and asymmetric dumbbell shape after Min oscillation decay on the membrane (Fig. 2, Supplementary Movie 4). This observation therefore suggests that, in these vesicles, the actomyosin bundles pinching the membrane are solely responsible for the sustained deformation obtained, and the observed constriction force (three orders of magnitude higher than the one Min

proteins could generate) is similar in nature to that characterized by Bashirzadeh et al.

However, after the stalling of myosin II on bundles occurred, contraction was arrested and deformation did not progress. It should be borne in mind that, in our system, myosin II also crosslinks actin and consumes ATP, an energy source employed by actin as well as MinD. Thus, we consider reasonable to suggest that under our encapsulating conditions with fascin as bundling agent, myosin II contracts the actin assembly until it reaches full condensate formation or stalls due to a compacted actin architecture or ATP shortage[28,66].

Enhancement of actin contraction is an important aspect that requires improvement, as our actomyosin architectures become stagnant over time and progression of the cytokinetic contraction is hampered. To effectively achieve a controlled contraction of the ring and its decrease in diameter on the membrane, there are a few approaches that could be implemented in our system. From the activation of latent myosin II with blebbistatin to trigger contraction late after encapsulation[67,68], to the assembly of a mix-polarity cortex remodelled by actin turnover (naturalistic route for synthetic division)[10,47,66], the use of this motor protein could render promising results. Alternatively, as Kučera et al. showed[13], an interesting element that could substitute or supplement myosin activity is anillin, a passive actin crosslinker capable of generating contractile forces. Regardless of the approach, future studies with optimized contractile toolsets could include the MinDE system to efficiently position these actin-based assemblies at mid-cell and deform lipid vesicles.

Collectively, the findings here provide a potential avenue for the equatorial localization of contractile actomyosin rings and the spatial targeting of mechanical forces at the vesicle membrane. Hence, being one step closer to a well-defined self-division of artificial minimal cells, future investigations should consider expanding the system reconstituted here with other division toolsets like a membrane expansion system to engineer membrane growth, an imperative to achieve sustained synthetic division.

### Towards the integration of different functional modules to build a synthetic cell

In conclusion, we show that positioning and confinement of contractile actomyosin rings in vesicles to a defined zone of future constriction, ideally in the equatorial region, is possible by employing the versatile MinDE protein system. We thus provide an example of a successful synthetic integration of functional toolkits from different organisms for the division of minimal vesicular systems. As synthetic biology strives to achieve the construction of an artificial cell, minimal modules reconstituted separately must be combined. Working towards the optimization of experimental conditions to meet the needs of all components may be the next challenge of the field. However, the efforts to interlace diverse functional modules might provide the field with interesting outcomes, like emergent protein functions or unexpected advantageous effects arising from the interplay of completely unrelated families of biomolecules.

## Methods
### Proteins
Actin (alpha-actin skeletal muscle, rabbit), ATTO647-Actin (alpha-actin skeletal muscle, rabbit) and Biotin-Actin (alpha-actin skeletal muscle, rabbit) were purchased from HYPERMOL (Germany). Myosin (rabbit muscle) was purchased from Cytoskeleton Inc (Tebubio GmbH, Offenbach, Germany). Fascin (human, recombinant) was purchased from Cytoskeleton Inc (Tebubio GmbH, Offenbach, Germany) and HYPERMOL (Germany). Stock solutions for all purchased cytoskeletal proteins were obtained by following the handling instructions of the manufacturer. Stock solutions of neutravidin (Thermo Fisher Scientific Inc., Massachusetts, USA, Cat# 31000) were prepared by dissolving the protein in water according to reconstitution instructions.

MinE-His, His-MinD and His-EGFP-MinD were purified as described in previous reports[69,70]. Briefly, His-tag carrying proteins were purified via affinity chromatography using Ni-NTA columns. Transformed *E.coli* BL21 cells were lysed by sonication and cell lysates were centrifuged to discard debris. Supernatants were loaded into Ni-NTA columns and proteins were eluted in storage buffer (50 mM HEPES, pH 7.25, 150 mM KCl, 0.1 mM EDTA, 1 mM TCEP, 10% Glycerin). Protein purity was confirmed via SDS-PAGE. Protein concentration and protein activity were determined via Bradford assay and ATPase assay, respectively.

## Crowder and density gradient solutions

BSA (Sigma-Aldrich, St. Louis, USA, Cat# A6003) stock solutions were prepared by dissolving the lyophilized powder in Millipore water at ~100 g/L as previously described[9]. To remove undesired debris from BSA solutions, three washing steps were performed with Millipore water using Amicon Ultra-0.5 centrifugal filters 50 kDa MWCO (Merck KGaA, Darmstadt, Germany). Concentration of final stock solutions (ranging from 170 to 300 g/l) were determined via Bradford Assays and stored at -20 °C. Ficoll70 (Sigma-Aldrich, St. Louis, USA, Cat# F2878) was dissolved in Millipore water and left at 4 °C in a rotary shaker for 24 h. Stock concentrations were calculated from the weight of Ficoll70 added and the final volume of solution obtained (625 g/L). Ficoll70 stocks were subsequently stored at -20 °C. 60% Iodixanol (OptiPrep™, Cat# D1556) was purchased from Sigma-Aldrich (St. Louis, USA).

## Vesicle production

All lipids were purchased from Avanti Polar Lipids (USA). For single-phase vesicles, a lipid-oil emulsion was prepared by dissolving 1-palmitoyl-2-oleoyl-sn-glycero-3phosphocholine (POPC), 1-palmitoyl-2-oleoyl-sn-glycero-3-phospho(1'-rac-glycerol) (POPG) and 1,2-dioleoyl-sn-glycero-3-phosphoethanolamine-N-(cap biotinyl) (sodium salt) (18:1 Biotinyl CAP PE) at a 6.9: 3: 0.1 molar ratio in 2.5 g/L final concentration and drying the mixture for 15 min under a nitrogen stream. In the case of experiments employing 1,2-dioleoyl-sn-glycero-3-phosphoethanolamine (DOPE), a mixture of POPC, POPG, Biotinyl CAP PE and DOPE were prepared in a 6: 3: 0.9: 0.1 molar ratio. For all lipid mixes, to ensure full evaporation of chloroform, mixtures were placed in a vacuum-sealed desiccator for at least 2 h. Inside a glove box (right before encapsulation experiments) 37.5 μL of decane (TCI Deutschland GmbH, Germany) was added to the dried lipid film and, once dissolved, 1 mL mineral oil (Carl Roth GmbH, Germany) was added and the lipid-oil suspension vigorously vortexed until a clear solution was obtained. For phase-separated vesicles, the lipid mix used was DOPC: DOPG: DPPC: DPPG: Chol (17.5: 7.5: 31.5: 13.5: 30) labeled with 0.001 mol% ATTO-655 DOPE binding to the Ld phase. The lipid mix (3.2 mM) was dissolved in chloroform and dried in a glass vial under $N_2$ flow for ~15 minutes. The dried film was then suspended in a mixture of decane (20 μL) and mineral oil (480 μL) and sonicated at elevated temperatures for ~30 minutes. Both single single-phase and phase-separated GUVs were produced with the double emulsion transfer method following a recently reported protocol for vesicle generation with purified proteins in 96 well-plates[9,71]. To ensure iso-osmolar conditions between the inside and outside of GUVs, the osmolarity of inner encapsulating solutions was measured with a osmometer (Fiske Micro-Osmometer model 120, Fiske Associates, Norwood, MA, USA) and outer glucose solutions with matching osmolarities were used as outer aqueous environment where GUVs are collected after production for subsequent imaging. The density of the inner mixture depends on the experimental conditions tested but to generate all the vesicles here reported we centrifuged well-plates at 600 × *g* for 10 min. In the case of single-phase vesicles centrifugation was performed at RT. For phase-separated GUVs, well-plates were centrifuged at 37 °C and the sample was allowed to cool to RT for ~30 min before imaging.

## Encapsulation of the system in single and double-phase GUVs

Depending on the experimental conditions tested, the final concentrations of the proteins varied but the procedure remained the same. All the steps were performed on ice except for the final encapsulation on 96 well-plates. First, we prepared a 35 μM actin mix (A-Mix) comprised of 86% G-actin, 10% ATTO647-actin and 4% biotinylated actin in water. Once we were ready to encapsulate, we prepared the following inner reaction mix: 4% OptiPrep™, 0.01 g/L Neutravidin, 10 g/L BSA, 10-50 g/L Ficoll70, 3-3.2 MinD (70% His-MinD, 30% EGDP-MinD), 1.6-3 MinE, 1.5-4 μM Actin (from the A-Mix), 0.3-2 μM fascin, 0-0.05 μM Myosin and 5 mM ATP (frozen stocks supplemented with 5 mM MgCl₂) in a final buffer concentration of 50 mM KCl, 10 mM Tris-HCl and 5 mM MgCl₂. To ensure no pre-polymerization, bundling or contraction of actin occurs prior encapsulation, actin, fascin, myosin and ATP were added in that order seconds before generating the vesicles via centrifugation.

## Fluorescence microscopy

Imaging of vesicles was performed on a LSM800 confocal laser scanning microscope using a C-Apochromat 40 × /1.2 water-immersion objective (Carl Zeiss, Germany). Fluorophores were excited using diode-pumped solid-state lasers: 488 nm (EGFP-MinD) and 640 nm (ATTO647-Actin).

## Image analysis

Processing and analysis of all acquired images were conducted using Fiji (v1.53f51)[72], a custom-written script in MATLAB (R2022a) and OriginPro (2021b), the latter also being used for plotting datasets. Z-Stacks were reconstructed in Fiji using the Standard Deviation Z-Projection. For kymographs, the Multi Kymograph function was used and ROIs were drawn manually following the GUVs' fluorescence intensity at their equatorial cross section with the free-hand selection, fitting them to splines before kymograph retrieval. For aspect ratio calculations, Fig. 2a, b and Supplementary Fig. 4a GUVs were traced by hand using the free-hand selection and major and minor axes were retrieved from Fiji measurements. For Fig. 2c, all time frames from the EGFP channel were processed first with a Gaussian filter of radius 1 and then Otsu-thresholded (frames where GUV contours where incorrectly thresholded were discarded). After filling holes, ROIs retrieved from the particle analyzer were fitted to an ellipse to measure the major and minor axes. Intensity profiles in Fig. 3b were obtained applying the Plot Profile Fiji function to segmented lines after being fitted to splines. For the waterfall 3D plot in Supplementary Fig. 5b, a 5 μm intensity line-plot at the bleb was drawn in Fiji to obtain fluorescence intensity values at six different time points. The data were plotted in Origin using the 3D waterfall plot. In Fig. 3d, curvatures were calculated by drawing ROI circles fitting the blebs and retrieving radii measurements from Fiji.

## Analysis and quantification of actomyosin phenotypes inside GUVs

For Fig. 1c, data for both fascin/actin molar ratios studied were obtained by acquiring Z-stacks of 3 separate experiments (triplicates for presence and absence of Min proteins) after Min oscillation decay for those where Min proteins were added. 150 GUVs of different sizes taken from these 3 experimental runs were pooled and analyzed together. Those presenting encapsulated actin bundles were classified into four categories (rings, asters, stiff-straight bundles and soft webs) by analyzing the acquired Z-stacks and their Standard Deviation Z-Projection. GUV diameters were obtained by manually drawing circle ROIs at the vesicle equator and vesicles were grouped into 4 categories according to their size. GUVs with no actin assemblies inside were not considered for the analysis. Frequencies were calculated from the number of vesicles belonging to each actin-assembly category and the total number of pooled GUVs analyzed per condition (150 vesicles). Datasets were normalized using the total number of vesicles analyzed

in each size group. For Supplementary Fig. 1a, three independent experiments were perfomed with a total of 336 GUVs analyzed per experimental run (GUVs analyzed per timepoint = 112). GUVs with a diameter above 25 μm were not considered for the analysis. To be rigorous on the time, 1 hour was considered as the first timepoint as it takes ~20 min for vesicle production and ~30 min to complete the acquisition of a high-resolution Tilescan in 3D. Samples were kept at 20 °C and later imaged at 7 and 24 h.

### Analysis of MinDE-induced folding of actin bundles

To quantify the angle between the two bundle ends attached to the vesicle membrane, the equator of the ATTO647-Actin channel was taken, and this time-series segmented using the "Moments" threshold method. To automatically detect both bundle ends as features and track their position over time, the Fiji plugin 'TrackMate' was employed[73]. The LoG detector was configured with threshold 7, radius 6 px, median filtering, and subpixel localization. Coordinates of the spots detected (two per time frame corresponding to the bundle ends) were obtained with TrackMate's Simple LAP tracker using the following settings: linking max distance 1 px, gap-closing max distance 15 px, gap-closing max frame gap 1. The angles between the two bundle ends over time were obtained with a MATLAB custom script taking spot coordinates as vector with the vesicle's center point as origin.

### Statistics & reproducibility

Sample sizes for vesicle classification were not predetermined prior to experiments. The sample size was chosen from a standard number relevant for the field and taken from at least three independent replicates (total number of GUVs analyzed per experimental run between 150 and 336, number specified in each figure caption). Employing this sample size for vesicle classification allowed actin morphologies to be clearly distinguished and classified with sufficient representation. Exclusion criteria was pre-established before data analysis. Vesicles containing two different actin phenotypes in their lumen and vesicles with encapsulated lipid or protein aggregates were excluded to ensure that only high quality GUV data was employed in this study. Empty vesicles were not taken into consideration for Fig. 1c but we analyzed their frequency in Supplementary Fig. 1b.

For actin phenotype classification in Fig. 1c, three independent experimental replicates were performed for each condition and GUVs from these replicates were pooled together for their analysis. For Supplementary Fig. 1b, three independent experimental runs were performed, with a total of 336 GUVs analyzed per experimental run (GUVs analyzed per timepoint = 112). All qualitative data in the form of microscopy images were replicated with at least ten independent experiments for each protein molar ratio specified. From these experiments ($n = 25$) diffusiophoresis, blebbing and membrane deformations were observed on many different vesicles from different experimental runs confirming their reproducibility. The vesicles with the most notable effects were then chosen for the analysis. For phase separation experiments, three independent control experiments were performed. For the encapsulation of actomyosin with Min proteins inside phase-separated vesicles, five independent encapsulation experiments were performed and similar results observed. Finally, the experiments were not randomized and the investigators were not blinded to allocation during experients and outcome assessment.

### Reporting summary

Further information on research design is available in the Nature Portfolio Reporting Summary linked to this article.

## Data availability

All data sets supporting the findings of this study are provided in the Source Data file. The Source Data have been deposited in the Figshare database under accession code: https://doi.org/10.6084/m9.figshare.26893822[74]. Due to large file size, the original acquired images and timelapse videos are freely available upon access request. This access can be obtained by contacting the corresponding author, who will aim to process requests within a week. Source data are provided with this paper.

## Code availability

The custom MATLAB script employed to calculate the folding angle of actin rings over time is available with full access in the GitHub repository: https://github.com/MariaReverteLopez/Ring-Folding-Analysis-inside-GUVs.

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

## Acknowledgements

The authors would like to thank the MPIB Core Facility for assistance in protein purification, Michaela Schaper for plasmid cloning, Kerstin Röhrl for protein purification, Sandra Ortmeier for lipid preparations and Sigrid Bauer for her advice on vesicle formation and unconditional scientific support. The authors would also like to thank Jan-Hagen Krohn for assistance in confocal microscopy, as well as Adrián Merino-Salomón and Shunshi Kohyama for helpful discussions on crowder conditions and protein encapsulation. For insightful discussions on the mechanism behind vesicle blebbing, the authors would like to thank Prof. Erwin Frey, Dr. Henri Franquelim, Dr. Ivan Maryshev and Henrik Weyer. This work was supported by the Deutsche Forschungsgemeinschaft (P.S. and M.J.). Y.Q. received funding from the European Union's Horizon 2020 research and innovation programme under the Marie Skłodowska-Curie grant agreement no. 859416. M.R.-L., Y.Q. and V.B are part of IMPRS-ML, and M.R.-L. is part of the ONE MUNICH Project supported by the Federal Ministry of Education and Research (BMBF) as well as the Free State of Bavaria under the Excellence Strategy of the Federal Government and the Länder. The authors would also like to acknowledge the support of the Center for Nanoscience (CeNS), Munich.

## Author contributions

M.R.-L. and P.S. conceived the study. M.R.-L. designed and performed encapsulation experiments, analyzed data and interpreted results. N.K and M.R.-L. designed and carried out phase-separation experiments. M.R.-L and Y.Q analyzed blebbing vesicles. V.B. assisted with the supplementary experiments. M.J. provided technical advice on protein conditions for encapsulation. M.R.-L. and P.S. wrote the manuscript and all authors revised and approved the final version of the manuscript.

## Funding

## Competing interests

The authors declare no competing interests.
