## [Transparent Peer Review file · Nature Communications]

Self-organized spatial targeting of contractile actomyosin rings for synthetic cell division

Corresponding Author: Professor Petra Schwille

Version 0:

Reviewer comments:

Reviewer #1

(Remarks to the Author)

The present manuscript by the Schwille group demonstrates the reconstitution of contractile actomyosin rings in GUVs. The key novelty of this paper lies in the combined use of actomyosin rings and the MinDE protein system to achieve self-organized spatial targeting of contractile structures within synthetic cells. This integration allows for the precise positioning of actomyosin rings at the equatorial plane of vesicles, enabling controlled membrane deformation and contributing to synthetic cell division. Although both, Min reconstitution and actomyosin reconstitution has been achieved previously by the same group, it was not clear that the two systems can act synergistically. As such I find the paper interesting and publishable. I have a few comments below.

- 1) Right now the system combines Eukaryotic and bacterial elements. Can the authors think about ways to make also the oscillatory machinery Eukaryotic?
- 2) The authors state that "To generate the tension required for ring constriction, actin must associate with myosin, the key motor protein that drives contractility of the actin assembly and induces furrow ingression". This, to my limited understanding, seems to be an outdated view. It has been shown that passive crosslinkers can contract actin rings in the absence of myosin (e.g. <https://www.nature.com/articles/s41467-021-24474-1>). Could similar approaches be interesting for the synthetic cell as well?
- 3) Figure 1c: The authors state $n=3$. Please include the error bars then. How exactly was the classification performed? Was it done manually? How crucial is time as a factor? Have the authors observed if the distributions of e.g. asters vs rings shift when the sample is observed at a different time point?
- 4) The authors state that myosin was added during the encapsulation process. How did they ensure constriction did not happen prematurely, i.e. during the process of vesicle assembly?
- 5) It would be really interesting to see what happens if membrane pores like alpha-hemolysin are included in the system. In this way, the effect of osmotic pressure can be excluded/minimized. At the same time, e.g. ATP can be resupplied from the bulk solution. The experiment should be doable, so I strongly suggest to give it a go.
- 6) Can the authors discuss the effect of membrane composition? It would be interesting to include e.g. cone-shaped lipids in the mix.
- 7) It is very interesting that the authors included phase-separated GUVs. It would be cool to see also completely phase separated GUVs (i.e. Janus GUVs). Did the authors try?

Reviewer #2

(Remarks to the Author)

The manuscript "Self-organized spatial targeting of contractile actomyosin rings for synthetic cell division" by Reverte-López et al. demonstrates MinDE-driven spatial positioning of actomyosin rings at the equatorial plane in vitro inside GUVs. The authors showed that MinDE can best transport actin bundles to the mid-cell region when doing pole-to-pole oscillations and the synergistic effect of MinDE oscillations and actomyosin bundle contraction can induce membrane deformation and bleb-like protrusions. Overall, the authors have done a commendable job characterizing this reconstitution system under various conditions and showcased the potential of integrating functional machinery from different organisms to achieve synthetic cell division, a critical step towards producing artificial cells. However, the extent to which the system deforms the membrane is small, and the interpretation of the membrane-shaping mechanisms is not definitive. The following suggestions are intended

to help make this manuscript acceptable:

Major concerns:

- 1) The authors observed four actin phenotypes in GUVs, with the fraction of each phenotype varying with the size of GUVs. The authors should explain how the size of GUV impacts actin architectures. Was the size of the GUV taken into consideration throughout the paper?
- 2) The mechanisms of the membrane deformation in this work require further investigation. The data suggest that deformation arises from actin bundle contraction and the formation of MinDE patches in actin filament-delimited regions. However, it is crucial to determine which is the dominant driving force. Membrane deformation by actin contraction or MinDE patches alone should be measured and compared to see which contributes more. This will provide a more comprehensive understanding of the system and its potential for synthetic cell division.
- 3) Although the combination of the spatial positioning tool MinDE and actomyosin contraction can bend the membrane, the induced furrow ingression is relatively small compared to the size of GUV in all tested conditions, far from achieving synthetic cell division. To address this, the authors could explore other factors that could enhance membrane deformation, such as altering the concentration of MinDE or actomyosin, or introducing other proteins that could interact with the system.

Minor points:

- 1) The MinD patch (green) in the cartoon of Figure 2 panel b was positioned opposite to its location in the corresponding fluorescent image.
- 2) In the Discussion, the authors used “generation of bilayer asymmetry due to the insertion of their alpha-helix at the inner leaflet” as one of the possible ways in which Min proteins might deform the membrane. Typically, the insertion of an alpha-helix into one leaflet of a lipid bilayer leads the membrane to bend towards the insertion leaflet, which is opposite to the bending direction driven by Min proteins in this manuscript.
- 3) Do Min proteins participate in liquid-liquid phase separation? Is the Min protein-rich region formed by protein phase separation? At least one study (Yuan et al., PNAS, 2021) has provided evidence for membrane bending driven by protein phase separation and demonstrated similar negative curvature generation by two-dimensional protein phase separation.
- 4) It is hard to interpret the “actin-filled membrane out-bud formed at the constriction site” from the data presented in Supplementary Movie 3 because it was already present in the movie’s first frame.

Reviewer #3

(Remarks to the Author)

Version 1:

Reviewer comments:

Reviewer #1

(Remarks to the Author)

The authors have done a considerable amount of new experiments that address my concerns.

Reviewer #2

(Remarks to the Author)

The authors have incorporated new measurements and interpretations to support their arguments. However, there are still some unresolved issues related to the membrane bending mechanisms that need further clarification. One issue is that, in some cases, the authors responded to our concerns in their rebuttal letter but did not address the issue in the revised paper. Another important issue is that they seem confused about the direction of membrane bending mediated by certain mechanisms.

1. In their response, the authors referred to several key studies comparing the contributions of actin bundles and Min proteins. These references and their corresponding data should be incorporated into the Discussion section to highlight the roles played by actin and Min proteins at various stages of membrane deformation. For example, statements in the response such as “forces exerted by Min proteins played a major role in the generation of these deformations in the first place,” “the expansive force induced by Min proteins is four orders of magnitude lower than actin,” and “the actomyosin bundles pinching the membrane are the sole responsible for the sustained deformation” should be more fully discussed in the manuscript.
2. Importantly, protein crowding drives membrane bending towards the leaflet of the lipid bilayer where the proteins reside (e.g., Stachowiak et al., Nature Cell Biology, 2012; Busch et al., Nature Communications, 2015). This is opposite to the membrane bending direction driven by Min proteins in this study. The authors do not understand the biophysical mechanism by which molecular crowding causes membranes to bend.
3. The authors list alpha-helix insertion, molecular crowding, and scaffolding as possible explanations for the membrane bending they observe. However, alpha-helix insertion into one leaflet of the lipid bilayer typically bends the membrane towards the insertion side, which is opposite to what they observe for Min proteins. If the authors wish to propose insertion as a mechanism for Min action, they should acknowledge the contradiction and explain how this would work.
4. The pressure generated by the clustering of ordered Min protein oligomers at the bilayer should also be mentioned in the context of membrane bending. This discussion could provide additional insights into the forces driving membrane deformation to offer a more biophysics-based understanding of the mechanism.

Reviewer #3

(Remarks to the Author)

Version 2:

Reviewer comments:

Reviewer #2

(Remarks to the Author)

We are satisfied with the revisions to this manuscript.

Reviewer #3

(Remarks to the Author)

First, we would like to thank all three reviewers for their meaningful feedback. Following their insightful suggestions, we carried out additional experiments and we now present these results in new supplementary figures (Supplementary Fig. 3, 4 and Supplementary Fig. 1a). In addition, we performed the corrections suggested by the reviewers and incorporated new explanations and descriptions to the main text according to their comments and questions. These changes (inclusions or deletions to the first submitted manuscript) are highlighted in yellow in the main text and supplementary information. In this report we address all the raised comments in blue in a point-by-point manner.

Reviewer #1 (Remarks to the Author):

The present manuscript by the Schwille group demonstrates the reconstitution of contractile actomyosin rings in GUVs. The key novelty of this paper lies in the combined use of actomyosin rings and the MinDE protein system to achieve self-organized spatial targeting of contractile structures within synthetic cells. This integration allows for the precise positioning of actomyosin rings at the equatorial plane of vesicles, enabling controlled membrane deformation and contributing to synthetic cell division. Although both, Min reconstitution and actomyosin reconstitution has been achieved previously by the same group, it was not clear that the two systems can act synergistically. As such I find the paper interesting and publishable. I have a few comments below.

We thank reviewer #1 for their positive comments and for suggesting additional experiments and very insightful ideas which have improved our understanding of the system and its interaction with the GUV membrane.

1) Right now the system combines Eukaryotic and bacterial elements. Can the authors think about ways to make also the oscillatory machinery Eukaryotic?

From the perspective of engineering a synthetic cell, this is a very interesting question. Similarly to the MinDE waves in *E. coli*, there exist other reaction-diffusion networks in eukaryotic cells which present wave forming or oscillatory pattern behaviour. Some examples are: (i) the ParABS system, a set of proteins involved in cell polarity and present in the *C. elegans* embryo (Goehring et al., Science, 2011); (ii) Cdc42, a GTPase from the Rho family in fission yeast which dynamically oscillates between the two poles due to positive and negative feedback loops controlled by activating and inhibiting factors (Das et al., Science, 2012); (iii) and Rho, a key GTPase for animal cell cytokinesis which shows dynamic cortical waves directed by its positive feedback mechanism of activation and the negative feedback exerted by F-actin polymerization (Landino et al., Current Biology, 2021).

Although these examples could be considered as a better strategy for their combination with a contractile actomyosin ring, the *in vitro* reconstitution of a eukaryotic-based oscillating apparatus presents many challenges. Firstly, there is still a lot of work needed to describe the molecular mechanisms behind their oscillations, given that different mechanistic models have been proposed for each of the aforementioned systems. Secondly, once the functional elements and their regulation are understood, we would need to find the conditions for the reconstitution of their oscillatory pattern *in vitro*, a feat still to be achieved. Finally, not only these systems involve many functional elements and coordinated high-order pathways for their spatiotemporal control, none of these systems have shown to have diffusiophoretic capabilities to reorganize minimal cytokinetic architectures bound to membranes, an essential feature required to reduce the complexity of our reconstituted system. Interestingly, Min

proteins have shown to form dynamic oscillations when expressed in yeast and mammalian cells, where they also spatiotemporally regulated proteins bound to their membrane, a feat that brings new possibilities for applications and the development of chimeric systems (Ramm et al., Nature Physics, 2021; Rajasekaran, et al., Cell, 2024). Thus, owing to its simplicity, diffusio-phoretic capabilities and extensive theoretical and experimental study, we consider the positioning MinDE system a feasible oscillatory machinery to rely on.

2) The authors state that “To generate the tension required for ring constriction, actin must associate with myosin, the key motor protein that drives contractility of the actin assembly and induces furrow ingression”. This, to my limited understanding, seems to be an outdated view. It has been shown that passive crosslinkers can contract actin rings in the absence of myosin (e.g. <https://www.nature.com/articles/s41467-021-24474-1>). Could similar approaches be interesting for the synthetic cell as well?

We agree with the reviewer that it is a very simplistic and outdated view and we apologize for failing to provide a more detailed description in the introduction. **We removed the referred sentence and the manuscript now contains a more complete explanation on the key players involved in cytokinetic ring formation and constriction** (page 3, line 49).

Indeed, as mentioned in the discussion, anillin is a very promising element that could be incorporated as a passive crosslinker to enhance ring contractility. Although a lot of work is still needed to find the right experimental conditions at which contraction is enhanced, we performed some preliminary encapsulation experiments with purified anillin, actin and the MinDE system showing promising results. More detailed information about these experiments can be found in the answer to reviewer #2, Q3, where we discuss the obtained results.

3) Figure 1c: The authors state $n=3$. Please include the error bars then. How exactly was the classification performed? Was it done manually? How crucial is time as a factor? Have the authors observed if the distributions of e.g. asters vs rings shift when the sample is observed at a different time point?

We apologize for the confusing wording employed for the figure caption. No statistical analysis was performed between replicates as the GUVs from all three experiments were pooled together to obtain a significant population number which could be classified according to size and actin phenotype. The data represented is therefore not an average of the three experimental runs. **We removed: “Experiments performed per condition $n = 3$ ” from the figure caption to prevent confusing the reader.**

Regarding the methodology behind classification, diameter measurements were done manually fitting circle ROIs in ImageJ due to GUVs being at different planes and ImageJ scripts failing at finding the right equatorial plane for many of them. Additionally, as we could not design an algorithm to effectively identify actin phenotypes, this classification was also performed manually by using the 3D reconstruction of the vesicles and by navigating through the 2D images acquired to ensure correct identification. **To provide with a complete explanation of this analysis, we included a detailed description in the Methods section** (page 24, line 549).

Time is indeed a factor which plays a role in our system. The more the sample is left at RT, the more effect Min oscillations have on actin bundles and the thicker and more numerous these bundles appear due to the ongoing polymerization and bundling of actin. **To study this distribution over time and answer the reviewer’s question, we performed the same**

experiment from Fig. 1c (employing 0.25 as fascin/actin molar ratio) three times, and we acquired images at different time points to analyze the phenotypes obtained. In this case, we performed statistical analysis on the experiments to show that our encapsulation process yield values close to the mean of the experimental runs performed. **We included the new bar plot obtained in Supplementary Fig. 1a and an example of sustained GUV deformation from actomyosin contraction at 24 hours in Supplementary Fig. 4a. We described these results in the main text in lines 151 (page 8) and 241 (page 11), and the methodology followed for the analysis is included in line 561 (page 25).** As it was not possible to represent the data like in Fig. 1c, due to the error bars precluding the proper plotting and reading of the stacked bins, we broke down the data according to phenotypes on the x-axis to facilitate plot interpretation. If the reviewer considers that statistical analysis for Fig. 1c is necessary, we can also take the data separately from the three replicates but, as mentioned, this would make impossible its representation and reading as the bar plots contain stacked bins for many data classes (Min presence, GUV size and actin phenotype).

4) The authors state that myosin was added during the encapsulation process. How did they ensure constriction did not happen prematurely, i.e. during the process of vesicle assembly?

We agree with this concern raised. To ensure that no pre-polymerization, bundling or contraction of actin happens while preparing the sample, we first mixed crowders, Min proteins and other constituents on ice. Seconds before encapsulation (once the lipid monolayer is ready), we promptly added actin, fascin, myosin and ATP in this order to the inner solution mix, with the Eppendorf placed always on ice. We then thoroughly mixed and quickly took 2,5 μ L from this inner solution mix to make the water in oil emulsion. As it takes 10 minutes to generate the GUVs from the w/o emulsion via centrifugation, polymerization and bundling could already start during this centrifugation step inside the vesicles. Unfortunately, we are not able to image the vesicle production process to report on the myosin effect at this time point. We always imaged our vesicles right after production and the hours following encapsulation, and the results here shown correspond to this time window.

Thus, **to provide the reader with a more detailed explanation of our encapsulation process and address this concern in the methods section, we added in line 525 (page 23) the following sentence:** "To ensure no pre-polymerization, bundling or contraction of actin occurs prior encapsulation, actin, fascin, myosin and ATP were added in that order seconds before generating the vesicle via centrifugation".

5) It would be really interesting to see what happens if membrane pores like alpha-hemolysin are included in the system. In this way, the effect of osmotic pressure can be excluded/minimized. At the same time, e.g. ATP can be resupplied from the bulk solution. The experiment should be doable, so I strongly suggest to give it a go.

We thank the reviewer for this meaningful suggestion. Indeed, adding pores to the membrane is a very interesting strategy to manipulate the osmotic pressure, inner content and membrane properties. **Following the insightful suggestion from the reviewer, to supplement our system with more ATP, we performed experiments with alpha-hemolysin pores (see figure on next page).** Due to difficulties with the delivery of purified alpha-hemolysin from several distributors, we employed PURE cell-free expression to obtain alpha-hemolysin which we could incorporate to our co-reconstituted MinDE actomyosin GUVs. We expressed the protein for 4 hours and centrifuged the resulting synthesized protein at 15000 rpm for 10 min to remove aggregates.

First, we tested the expressed alpha-hemolysin in GUVs containing our standard lipid and crowder compositions (refer to Fig. 3a for detailed crowder solution mix) by studying the

leakage of ATTO647 dye after alpha-hemolysin addition (panel a from figure below). After we confirmed that the dye leaked and GUVs lost brightfield contrast similarly to other studies found in literature (M. Thomas et al., *Chem. Commun.*, 2017; Van de Cauter et al., *Synth. Biol.*, 2021), we incorporated the same amount of alpha-hemolysin to GUVs containing our co-reconstituted system. The outer solution in this case containing glucose, 5 mM ATP and salt concentration matching inner buffer content.

On the one hand, as pores started forming at the membrane and brightfield images showed loss of contrast, we observed a change in MinDE oscillations (panel c). After 15-20 minutes, dynamic MinDE patterns inside our actomyosin vesicles transitioned to mostly circling oscillations (54%) and pulsing patterns (15%), while 23% of the GUVs imaged presented no oscillations, and only 8% showed the desired pole-to-pole pattern (GUVs analyzed = 26). This unfavourable oscillation transition might be due to changes in MinDE concentrations at the membrane, as the pores might be interfering negatively with their reaction-diffusion dynamics. Thus, we consider this pore-addition strategy not entire compatible with MinDE oscillations.

On the other hand, with the supplemented 5 mM ATP added from the outside, we did not observe an improved contraction of our actomyosin architectures after more than 1 hour of imaging. Actin architectures either remained the same, or collapsed into asters which also caused the bursting of the vesicles (panel b). With the experiments here performed no controlled web or ring closure was observed. This suggest that myofilaments might be limited by the coarsening of actin architectures (the fragmentation and compaction in the form of asters and thick bundles) and an addition of G-actin molecules and increase in actin-turn over might be necessary to remodel the actin structures inside the vesicles over time (Vogel et al., *eLife*, 2013; Sonal et al., *Journal of Cell Science*, 2019).

Overall, although this suggestion allowed us to supply more ATP from the bulk solution, the pores formed interfered with pole-to-pole MinDE oscillations rendering this strategy incompatible with their required positioning functionality. However, we consider the MinDE oscillation transition observed an interesting phenomenon which we will continue to study to yield this strategy compatible with MinDE patterns.

6) Can the authors discuss the effect of membrane composition? It would be interesting to include e.g. cone-shaped lipids in the mix.

We thank the reviewer for the insightful suggestion. Indeed, previous studies have shown that the composition and charge of the lipid membrane affect MinDE oscillations (Vecchiareli et al., Mol. Microbiology 2014; Kohyama et al., 2019, eLife). For example, MinD has higher affinity to anionic phospholipids, such as phosphatidylglycerol (PG) and cardiolipin (CL), compared to zwitterionic phospholipids like phosphatidylethanolamine (PE). The lipid composition thus modulates MinD binding to the membrane and its interaction with MinE, ultimately affecting reaction dynamics and their overall oscillatory behaviour.

Although membrane composition impacts the MinDE patterns obtained, **we followed the interesting idea presented by the reviewer and we performed experiments at increasing molar concentration of DOPE**, a lipid known for its inverse-cone shape. With our encapsulating protocol, 10% is the highest molar ratio we could incorporate to our membranes, as the GUV yield dramatically decreases above that percentage. Interestingly, at this high molar ratio, MinDE oscillations are not affected and dynamic patterns also arise. Moreover, besides highly curved and strange-shaped vesicles (see figure below), we observed membrane deformations like equatorial furrowing and blebbing in vesicles presenting actomyosin architectures. Although similar to the phenomena reported with just POPC-POPG vesicles, the fact that DOPE incorporation did not disturb MinDE oscillations is an interesting step to further build more membrane complexity together with other curvature-inducing lipids. **Thus, we have included these experiments in the new Supplementary Fig. 4b and presented the results in the main text (page 16, line 339).**

7) It is very interesting that the authors included phase-separated GUVs. It would be cool to see also completely phase separated GUVs (i.e. Janus GUVs). Did the authors try?

We thank the reviewer for his/her positive comment. Regarding the generation of Janus GUVs, **following the reviewer's suggestion, we tried to generate Janus vesicles containing our actomyosin-MinDE system.** Although it is possible to generate this type of phase-separated vesicles with electroformation, in this study we are dealing with a one-pot reaction in which encapsulation of proteins is required. Therefore, we attempted to adapt our double emulsion transfer protocol for the generation of Janus GUVs. Unfortunately, to obtain Janus vesicles

via double emulsion, centrifugation at high temperatures (>50 °C) is required to reach the melting temperature of the lipids (Baumgart et al., Nature, 2003; Dreher et al., Angewandte Chemie, 2021). This imposed a technical difficulty given that all our benchtop centrifuges could only reach 40 °C. In the current setup, we could observe a clear phase separation but with small patches of Lo domains because the experimental temperature (37 °C) was not high enough to coalesce the domains to yield Janus GUVs. Moreover, we consider very unlikely that proteins can still display functional behaviour after being subjected to such high temperature over a long period of time (at least 10 minutes for the vesicle production inside the centrifuge). Thus, with our current means, unfortunately, we cannot provide with an answer to the behaviour of the system in this type of phase-separated system.

Reviewer #2 (Remarks to the Author):

The manuscript "Self-organized spatial targeting of contractile actomyosin rings for synthetic cell division" by Reverte-López et al. demonstrates MinDE-driven spatial positioning of actomyosin rings at the equatorial plane in vitro inside GUVs. The authors showed that MinDE can best transport actin bundles to the mid-cell region when doing pole-to-pole oscillations and the synergistic effect of MinDE oscillations and actomyosin bundle contraction can induce membrane deformation and bleb-like protrusions. Overall, the authors have done a commendable job characterizing this reconstitution system under various conditions and showcased the potential of integrating functional machinery from different organisms to achieve synthetic cell division, a critical step towards producing artificial cells. However, the extent to which the system deforms the membrane is small, and the interpretation of the membrane-shaping mechanisms is not definitive. The following suggestions are intended to help make this manuscript acceptable:

We are very thankful to the reviewer for the valuable feedback and suggestions. While we agree that the contraction observed is of similar nature from the one obtained in previous reports, we consider necessary to remark that the main claim of our study is the co-reconstitution of two contractile and positioning synthetic modules and the successful positioning of eukaryotic-based contractile forces in minimal cells. A feat which can now be exploited in the actomyosin-driven synthetic division approach. An improvement of the system's deformation extent was certainly not our main goal. To emphasize that this is an aspect that requires of further improvement, **we changed the discussion (page 19, line 428) to avoid any misleading conceptualization of our results.**

Regarding the interpretation of the membrane-shaping mechanism, we described and explained the MinDE-driven blebbing observed based on previous reports addressing the force Min proteins induce on membranes. **We now provide a more extensive description of the mechanisms that might take effect and reference additional sources to support our hypothesis.** We consider that a further analysis of the precise molecular mechanism behind MinDE-driven membrane deformations would draw attention away from the main message of the present manuscript and belongs to a different type of study in which the focus is the theoretical characterization of MinDE-membrane interactions at the nanoscale. We will elaborate further on this matter in the point-by-point answers but we hope that the revised version of the manuscript and our answers will now satisfy the reviewer.

Major concerns:

1) The authors observed four actin phenotypes in GUVs, with the fraction of each phenotype varying with the size of GUVs. The authors should explain how the size of GUV impacts actin architectures. Was the size of the GUV taken into consideration throughout the paper?

We apologize to the reviewer for omitting this explanation in the results section. Indeed, the size of the GUVs generated impacts the type of actin phenotype and we took this into consideration in Fig. 1c. This impact is caused due to the persistence length of the actin filaments and the fact that actin bundles minimize their elastic energy by attaining the smallest curvature at the boundary. Based on the thorough description of this physical principle by previous studies (Miyazaki et al., Nat Cell Biol 17, 2015; Bashirzadeh et al., Commun Biol, 2021), **we added a short explanation of these findings in line 138 (page 7) to provide the reader with the context behind the experiments performed in Fig. 1c.**

Therefore, for the rest of the phenomena reported in our study, the size of the vesicles impacted indirectly the equatorial pinching/contraction and blebbing of the membrane, as these two effects are only observed when rings and webs form at the membrane. This observation is not surprising given that rings and soft bundle webs are the two actin phenotypes bound to the membrane that could efficiently induce equatorial furrowing and be reorganized by MinDE proteins. This observation is described across result sections 2.3-2.4. As an example, the figure below provides further evidence that we observed blebbing at different vesicle sizes (inner vesicle content as Fig. 3b). As expected, the acquisition of blebbing events is more difficult below 15 μm due to the fact that the frequency of soft web formation at this diameter is 20%.

2) The mechanisms of the membrane deformation in this work require further investigation. The data suggest that deformation arises from actin bundle contraction and the formation of MinDE patches in actin filament-delimited regions. However, it is crucial to determine which is the dominant driving force. Membrane deformation by actin contraction or MinDE patches alone should be measured and compared to see which contributes more. This will provide a more comprehensive understanding of the system and its potential for synthetic cell division.

While we agree with the reviewer that a measurement and comparison of both forces would provide a more comprehensive understanding of the system, this suggestion is very complicated to address with our means at the moment for many reasons. Firstly, GUVs are not the most suitable system to study these simultaneous forces. Blebbing is a very dynamic phenomenon that occurs in the span of seconds, which complicates its imaging and analysis

in 3D. For example, although segmentation of the 3D confocal images would allow us to determine the surface and curvature changes of the blebs, the voxel size of our 3D images is too big to provide a meaningful analytical estimate given that vesicles were imaged with very few Z-slices and a considerable inter-distance between slices ($\sim 2 \mu\text{m}$) to be able to track these very fast transformations in the entire 3D volume. However, if we were to study this phenomenon in a different system e.g.: free-standing membranes like black-lipid membranes or pore spanning membranes, these experimental set-ups would not show actin soft webs or rings, as these architectures appear under spherical confinement.

Secondly, it is not possible to co-reconstitute these two protein systems and, at the same time, measure their forces independently. In our control experiments, blebs are not observed in the absence of Min oscillations and Min proteins do not generate blebs in vesicles lacking inner actomyosin networks, we only observed bleb growth when actomyosin bundles delimited the GUV into different surface areas. Additionally, simultaneous to the development of this outward membrane deformations, the bundles acting as lateral diffusion barriers are displaced and reorganized due to MinDE diffusiophoretic transport at the membrane (Fig. 3b), causing the patches to constantly change in surface area. This dynamic reorganization and the new layer of complexity it adds to the system thus prevents us from studying the membrane properties at these patches and infer the forces acting on them.

However, several studies we referenced have already estimated the changes in membrane properties and the forces these two systems can generate on lipid vesicles separately. On one side, biotin-bound actomyosin bundles might affect lipid packing density in vesicles and increase bilayer rigidity and viscosity as a result of the myosin-induced contractile stress on the inner leaflet (Ioannou et al., JACS Au, 2024). Moreover, Bashirzadeh et al., calculated the force (in the piconewton range) that actomyosin bundles generate on GUVs to induce asymmetric dumbbell deformations similar to the ones we observed in Fig. 2c (Bashirzadeh et al., iScience, 2022). Given that we observed an analogous sustainable deformation after MinDE oscillation decay (Supplementary Movie 4), it is reasonable to suggest that, in this case, the actomyosin bundles pinching the membrane are the sole responsible for the sustained deformation and the constriction force in this vesicle is of similar nature to the one Bashirzadeh et al. characterized.

On the other side, it has been shown that Min proteins can induce mechanical forces and change the properties of lipid membranes upon their binding (Mazor et al., Biochimica et Biophysica Acta, 2008). To describe the mechanical force that MinDE oscillations can exert on free-standing membranes, Fu et al. employed free-standing membrane tube networks and flat vesicles (model membrane systems which support large-scale changes in membrane morphology) and quantified their Min-induced spreading dynamics and the forces at play. In the piconewton range, but four orders of magnitude lower than actin, Fu et al. calculated the expansive force induced by Min proteins without any lateral diffusive barriers, evidencing the mechanical work performed by these two proteins via ATP consumption (Fu et al., Angewandte Chemie, 2021). In our case, the progressive increase of EGFP-MinD fluorescence intensity at the membrane patches (Supplementary Fig. 5a,b) and the simultaneous development and growth of blebs suggest that the forces exerted by Min proteins played a major role in the generation of these deformations in the first place. We therefore included a discussion of these results in our manuscript based on these observed effects.

We hope our clarification convinced the reviewer of the experimental difficulties we face to fulfil his/her suggestion, but we are open to new ideas for possible experimental set-ups that could address this matter in future studies.

3) Although the combination of the spatial positioning tool MinDE and actomyosin contraction can bend the membrane, the induced furrow ingression is relatively small compared to the size of GUV in all tested conditions, far from achieving synthetic cell division. To address this, the authors could explore other factors that could enhance membrane deformation, such as altering the concentration of MinDE or actomyosin, or introducing other proteins that could interact with the system.

We are thankful for the suggestions and we agree with the reviewer that a significant enhancement of furrow ingression will be necessary to achieve synthetic division. Although showing this increase in contractile power was not among our main messages or goals for the present manuscript, **we followed the reviewer's suggestions and performed a series of experiments to explore new elements that could be incorporated to our positioning MinDE machinery and deform our GUVs further.**

First, to test if a different actin crosslinker could enhanced actin contractility, we performed experiments employing α -actinin and filamin at different molar ratios (see figure below). From these experiments we observed that, on the one hand, GUVs with filamin-bundled actin and myosin contain mostly asters on their membrane and lumen, with very few vesicles presented soft webs or rings that could be positioned by Mins. On the other hand, we did not observe GUV deformations when employing α -actinin alone. This crosslinker, in particular, yielded a very low number of actin architectures in GUVs and presented very thin web structures after a few hours of incubation for polymerization/bundling. Although previous reports employed a mix of different crosslinker to study their combined effect, we consider this strategy beyond the scope of the present study, given that this would unnecessarily increase the complexity of a system that was not meant to show a superior furrow ingression.

EGFP-MinD/Atto647-Actin

(3D, Z-projections)

Regarding the variation of MinDE concentrations, while different MinD/MinE molar ratios yielded similar results (see Supplementary Fig. 2b), changing MinDE concentration could be detrimental (see figure below). Although lower concentrations still allow the generation of dynamic oscillations inside GUVs (Kohyama et al., Nat Comm, 2022), increasing their concentration above 4 μ M can induce the formation of quasi-stationary patterns on the GUV inner leaflet (like inverse spots and large meshes) due to the crowder composition added inside our vesicles (Reverte-López et al., Small Methods, 2023).

MinD-MinE (6-6 μM)

EGFP-MinD/Atto647-Actin
(3D, Z-projection)

EGFP-MinD/Atto647-Actin
(3D, Z-projection)

EGFP-MinD/Atto647-Actin
(2D, equator)

Finally, motivated by the inspiring work by Kučera et al., we substituted myosin in our inner vesicle mix with anillin. This non-motor crosslinker, able to propel the contractility of actin bundles without energy consumption, presents itself as a promising element which could induce a better network contraction inside our GUVs. Indeed, from our preliminary experiments, we observed that Min proteins were able to position anillo-actin rings and networks inside vesicles via their pole-to-pole oscillations (see figure below, lower panel), confirming once again MinDE robustness.

EGFP-MinD/Atto647-Actin
(3D, Z-projections)

However, the anillo-actin architectures obtained reached an equilibrium soon after we started imaging them, and the contraction we observed had stalled. Interestingly, many membrane remnants appeared floating on the glass (upper right panel), which suggest that GUVs were exploding from the contraction soon after vesicles were generated by double emulsion. Nevertheless, these preliminary experiments show that anillin is a promising element that could work in combination with myosin to provide the system with a new source of contractile

power. Although the conditions here employed allow the formation and positioning of anillo-actin networks, further work is required to explore their parameter space to find the right conditions that could allow the controlled contraction of these architectures and accomplish synthetic division.

Given that the anillin-driven furrow ingression shown here is similar to the one presented in Fig. 2b-2c, we did not add these results to the main text as they do not modify the main message of the study in a relevant way.

Minor points:

1) The MinD patch (green) in the cartoon of Figure 2 panel b was positioned opposite to its location in the corresponding fluorescent image.

We thank the reviewer for the detailed correction of our manuscript. **We changed the cartoon as suggested.**

2) In the Discussion, the authors used “generation of bilayer asymmetry due to the insertion of their alpha-helix at the inner leaflet” as one of the possible ways in which Min proteins might deform the membrane. Typically, the insertion of an alpha-helix into one leaflet of a lipid bilayer leads the membrane to bend towards the insertion leaflet, which is opposite to the bending direction driven by Min proteins in this manuscript

Firstly, we would like to apologize for the unclear exposition of our hypothesis in the manuscript. The reviewer is right, in our discussion we propose several mechanisms behind the membrane deformation observed, being the MTS domain insertion one of them; a phenomenon which indeed would create an inverse local curvature via the so-called “wedging effect” (Shih et al., PLOS ONE, 2011). However, we also refer in the next sentence to the possible scaffolding or crowding effects taking place which could answer this incongruity (Lipowsky, Advanced Biology, 2022; Derganc and Čopič, BBA biomembranes, 2016). This is because, given the complexity behind the characterization of this phenomenon, it is likely that several mechanisms take effect simultaneously and the nanoscale topological disturbance the MTS domain generates on the membrane cannot fully describe the deformation we observed.

To address this, we further hypothesized that the outward deformations observed arise as a result of crowding or scaffolding effects. In this case, an additional pressure with normal component, generated by the clustering of ordered oligomers of MinD at the bilayer patches, could be a plausible explanation for the curvature observed. As pointed out in the next question, this positive (outward) curvature would resemble the wetting effect induced by high-order molecular structures like condensates which create steric pressure at the bilayer (Su et al., Nat. Chem. 16, 2024). Nevertheless, as further theoretical work is needed to describe the molecular interactions at the membrane and explain this phenomenon in detail, we considered necessary to provide several possible answers for the effect observed based on already published studies (increased viscosity, MTS insertion and crowding/scaffolding effect), irrespective of the curvature generated. We will continue to look at this fascinating phenomenon in detail and report our findings in future studies.

To further clarify our hypothesis, we modified the discussion adding the term “crowding” and we now provide a short explanation based on the answer here given (page 18, line 403). In addition, we included two new references that support MinD oligomerization and the generation of outwardly-curved membranes via crowding effects.

3) Do Min proteins participate in liquid-liquid phase separation? Is the Min protein-rich region formed by protein phase separation? At least one study (Yuan et al., PNAS, 2021) has provided evidence for membrane bending driven by protein phase separation and demonstrated similar negative curvature generation by two-dimensional protein phase separation.

Theoretical and experimental studies have extensively described the reaction-diffusion mechanism behind MinDE pattern formation at the molecular level. As far as we know, apart from a few mechanistic aspects still unresolved, these studies do not present direct evidence that could suggest that Min proteins participate in liquid-liquid phase separation (Ramm et al., Cell Mol Life Sci, 2019). The “protein-rich region” corresponds to the area Min molecules bind to as they oscillate. Moreover, unrelated *in vitro* experiments performed in bulk and supported lipid bilayers in our lab did not show any phase separation when varying salt and crowder concentrations. As mentioned in our previous answer, we therefore speculate that the pressure arising from the cooperative binding of MinD oligomers to this area might be behind the curvature and deformation observed.

However, it should be borne in mind that phase-separation is a phenomenon observed in other pattern-forming systems. This is the case of the DNA-binding ParABS system, which previous report showed the formation of liquid condensates. However, this example, more than supporting the idea that Min proteins could phase-separate, highlights the differences in the molecular mechanisms governing pattern forming systems besides their parallel functional and modular elements. For a more detailed comparison of the two systems, we refer the reviewer to Merino-Salomón et al., Current Opinion in Cell Biology, 2021.

4) It is hard to interpret the “actin-filled membrane out-bud formed at the constriction site” from the data presented in Supplementary Movie 3 because it was already present in the movie’s first frame.

We apologize to the reviewer for the deficient presentation and description of this result. The bud formation process was not captured in real time due to the GUV already presenting the bud when our timelapse was started. However, we find reasonable to describe it as an actin-filled out-bud similar to the one observed by Litschel et al., Nat. Comm. 2021 for three reasons: i) the analysis of the bud’s cross-section revealed that it is attached to the vesicle membrane at the region where the actin bundle network is being positioned by Min proteins; ii) its 3D geometry, closer to a comma-like shape than spherical, rules out the possibility of this bud being a smaller GUV; iii) its movement over time. From timestamp 16:00 to 19:00 minutes in Supplementary Movie 3, Min proteins reorganize the bundle network and thus the bud move accordingly in a downward motion.

Although these membrane buds have already been reported, we considered appropriate to describe the bud observed and reference the study which described their generation. We are sorry if this description was taken as misleading given that we did not record its formation in real time. **We now modified the main text (page 11, line 230) and the video caption to notify the reader about this aspect. In addition, we included in the new Supplementary Fig. 3 a closer look at the vesicle cross-section and the bud to facilitate the interpretation of this observation to the reader.** We hope this clarifies the point raised by the reviewer. However, if the reviewer thinks an omission of the description of this phenomenon is necessary, we would remove it and adjust the manuscript accordingly.

Reviewer #3 (Remarks to the Author):

We thank the reviewer for his/her feedback and corrections.

We would like to thank the reviewers again for their insightful feedback, which has significantly improved our manuscript. Following the comments of reviewer #2, we have modified the discussion according to their suggestions and changed Figure 3c to illustrate our hypothesis. The changes and newly added paragraphs are highlighted in yellow in the main text. In this report, we will respond to all the comments raised in a point-by-point manner.

Reviewer #1 (Remarks to the Author):

The authors have done a considerable amount of new experiments that address my concerns.

We would like to thank the reviewer again for their feedback.

Reviewer #2 (Remarks to the Author):

The authors have incorporated new measurements and interpretations to support their arguments. However, there are still some unresolved issues related to the membrane bending mechanisms that need further clarification. One issue is that, in some cases, the authors responded to our concerns in their rebuttal letter but did not address the issue in the revised paper. Another important issue is that they seem confused about the direction of membrane bending mediated by certain mechanisms.

We are grateful for the reviewer's comments and suggestions. Their contributions were instrumental in improving the discussion section. We realized that our discussion lacked clarity and a more detailed explanation and emphasis on the aspects that distinguish Min reaction-diffusion dynamics from other membrane-binding protein systems. We have considered all the suggestions and made the appropriate changes to the Discussion (see responses below). We hope that the newly added explanations are clear and unambiguous and address all the reviewer's suggestions.

1. In their response, the authors referred to several key studies comparing the contributions of actin bundles and Min proteins. These references and their corresponding data should be incorporated into the Discussion section to highlight the roles played by actin and Min proteins at various stages of membrane deformation. For example, statements in the response such as "forces exerted by Min proteins played a major role in the generation of these deformations in the first place," "the expansive force induced by Min proteins is four orders of magnitude lower than actin," and "the actomyosin bundles pinching the membrane are the sole responsible for the sustained deformation" should be more fully discussed in the manuscript.

We agree with the reviewer and would like to apologize for not including the information provided in the rebuttal after the previous revision. The newly revised version of the manuscript now contains all the comments and references mentioned by the reviewer in the Discussion section, as detailed below:

For the comment "Forces exerted by Min proteins played a major role in the generation of these deformations in the first place", we wrote "**In contrast to bleb-like morphologies (Figs. 3 and 4), where forces arising from Min oscillations were essential to generate these deformations (Supplementary Fig. 6), in this study we also showed vesicles that exhibited sustained equatorial constriction and asymmetric dumbbell shape after Min oscillation decay on the membrane (Fig. 2, Supplementary Movie 4).**" (page 20, line 441).

For "the expansive force induced by Min proteins is four orders of magnitude lower than actin," and "the actomyosin bundles pinching the membrane are the sole responsible for the sustained

deformation” the new version of the manuscript now specifically refers to the pN forces that each system can generate and we cite the studies where these forces were calculated on **page 19 (line 413)** and **page 20 (line 439)**. We compare these forces and discuss their role on **page 20 line 445**: **“This observation therefore suggests that, in these vesicles, the actomyosin bundles pinching the membrane are solely responsible for the sustained deformation obtained, and the observed constriction force (three orders of magnitude higher than the one Min proteins could generate) is similar in nature to that characterized by Bashirzadeh *et al.*”** We would also like to apologize for an erratum included in the previous revision rebuttal. It is three and not four orders of magnitude higher than Min-induced forces.

2. Importantly, protein crowding drives membrane bending towards the leaflet of the lipid bilayer where the proteins reside (e.g., Stachowiak *et al.*, *Nature Cell Biology*, 2012; Busch *et al.*, *Nature Communications*, 2015). This is opposite to the membrane bending direction driven by Min proteins in this study. The authors do not understand the biophysical mechanism by which molecular crowding causes membranes to bend.

We apologize to the reviewer for being unprecise in referring to the effect we observed as crowding. We wanted to draw attention to the high density binding of MinDE to the actin-delimited patches. However, as pointed out by the reviewer, steric pressure due to crowding effects on membranes has been shown to result in opposite bending direction to the one we observed. We were familiar with the work suggested by the reviewer and agree that simple crowding cannot account for the effect we are observing.

However, we would also like to emphasize that, in contrast to a simple accumulation of proteins on the membrane, we are dealing with a reaction-diffusion system. The non-equilibrium Min patterns exhibit complex dynamics resulting from reactive MinD-MinE fluxes with membrane-bulk coupling through ATP consumption. In contrast to the proteins used by Stachowiak *et al.* and Busch *et al.*, Min proteins exhibit permanent protein exchange between bulk and membrane as a consequence of their ATP-driven dynamic binding and dissociation. Therefore, in contrast to lateral pressure asymmetry on the membrane leaflets due to crowding, pressure may be directly exerted onto the membrane with an outward vector.

Importantly, previous studies under similar experimental conditions have shown that MinD binding alone cannot generate large-scale membrane deformations such as invaginations. The transient binding of MinE and the supply of ATP are required to generate deformations of free-standing membranes (Godino *et al.*, *Nature Communications*, 2019; Fu *et al.*, *Angewandte*, 2021). Therefore, although the bending direction observed here is in principle not energetically favourable (as suggested by Stachowiak *et al.*, proteins have reduced binding sites at the inner leaflet of the bleb), these results suggest that we are dealing with an energetically driven chemo-mechanical process for shape remodelling.

Exactly how Mins generate the observed highly curved blebs is a fascinating question that we cannot yet answer. Elucidating the exact molecular mechanism behind these deformations is not trivial, as it requires new structural insights. We would need to perform simulations and these would require the combination of several models, some of whose reaction terms and boundary conditions are still unknown (Fu *et al.*, *Nature Physics*, 2023). Indeed, some of the parameters defining the MinDE reaction-diffusion model are still to be determined and could be key to explaining the phenomenon at hand (Miyagi *et al.*, *Nano Letters*, 2018; Ye *et al.*, *Journal of the American Chemical Society*, 2018).

In short, to address this concern, **we have removed the term crowding from the new version of the manuscript and now provide a more detailed hypothesis for the deformations in the context of the Min reaction dynamics at the membrane, with additional references to relevant studies.** In particular, we also include the concept of membrane undulation. This is an aspect that may need to be taken into account when analyzing the changes in bleb surface area. In addition, to facilitate the interpretation of our hypothesis, **we have modified Fig. 3c by adding the direction of the force to the scheme. The three new paragraphs in the Discussion describing this hypothesis can be found on pages 18 and 19 from line 395 to 429.**

3. The authors list alpha-helix insertion, molecular crowding, and scaffolding as possible explanations for the membrane bending they observe. However, alpha-helix insertion into one leaflet of the lipid bilayer typically bends the membrane towards the insertion side, which is opposite to what they observe for min proteins. If the authors wish to propose insertion as a mechanism for Min action, they should acknowledge the contradiction and explain how this would work.

We apologize to the reviewer for the lack of clarity on this point. Our aim was to inform the reader about previous studies reporting the change in spontaneous curvature upon Min alpha-helix insertion, but as the reviewer correctly points out, this effect alone would contradict our results. The new version of the manuscript now mentions that our current knowledge of the effect of Min proteins on membrane properties and topology does not seem sufficient to describe our deformations and does not explain the membrane curvature we observed: **“Although previous reports have indicated the ability of Min proteins to increase membrane viscosity and change membrane topology by inserting their alpha helix, these two effects would not explain the membrane curvature observed here.” (page 18, line 404).**

4. The pressure generated by the clustering of ordered Min protein oligomers at the bilayer should also be mentioned in the context of membrane bending. This discussion could provide additional insights into the forces driving membrane deformation to offer a more biophysics-based understanding of the mechanism.

We thank the reviewer for the suggestion. **We have modified the discussion and the manuscript now mentions the clustering of MinD oligomers as a possible effect contributing to the deformations (page 18, line 407).** In addition, to avoid any misleading interpretation of our reasoning, we have indicated that the intrinsic curvature of these oligomers is not known.

Reviewer #3 (Remarks to the Author):

I co-reviewed this manuscript with one of the reviewers who provided the listed reports. This is part of the Nature Communications initiative to facilitate training in peer review and to provide appropriate recognition for Early Career Researchers who co-review manuscripts

We thank again the reviewer for their work on our manuscript.